# Targeting *mir128-3p* alleviates myocardial insulin resistance and prevents ischemia-induced heart failure

Andrea Ruiz-Velasco[1], Min Zi[1], Susanne S Hille[2,3], Tayyiba Azam[1], Namrita Kaur[1], Juwei Jiang[1], Binh Nguyen[1], Karolina Sekeres[4], Pablo Binder[1], Lucy Collins[1], Fay Pu[5], Han Xiao[6], Kaomei Guan[4], Norbert Frey[2,3], Elizabeth J Cartwright[1], Oliver J Müller[2,3]\*, Xin Wang[1]\*, Wei Liu[1]\*

[1]Faculty of Biology, Medicine, and Health, the University of Manchester, Manchester, United Kingdom; [2]Department of Internal Medicine III, University of Kiel, Kiel, Germany; [3]DZHK, German Centre for Cardiovascular Research, Partner Site Hamburg/Kiel/Lübeck, Kiel, Germany; [4]Institute of Pharmacology and Toxicology, Faculty of Medicine Carl Gustav Carus, Technische Universitaet Dresden, Dresden, Germany; [5]Edinburgh University Medical School, Edinburgh, United Kingdom; [6]Institute of Vascular Medicine, Peking University, Beijing, China

\*For correspondence:
oliver.mueller@uksh.de (OJM);
xin.wang@manchester.ac.uk (XW);
wei.liu@manchester.ac.uk (WL)

**Competing interests:** The authors declare that no competing interests exist.

**Abstract** Myocardial insulin resistance contributes to heart failure in response to pathological stresses, therefore, a therapeutic strategy to maintain cardiac insulin pathways requires further investigation. We demonstrated that insulin receptor substrate 1 (IRS1) was reduced in failing mouse hearts post-myocardial infarction (MI) and failing human hearts. The mice manifesting severe cardiac dysfunction post-MI displayed elevated *mir128-3p* in the myocardium. Ischemia-upregulated *mir128-3p* promoted *Irs1* degradation. Using rat cardiomyocytes and human-induced pluripotent stem cell-derived cardiomyocytes, we elucidated that mitogen-activated protein kinase 7 (MAPK7, also known as ERK5)-mediated CCAAT/enhancer-binding protein beta (CEBPβ) transcriptionally represses *mir128-3p* under hypoxia. Therapeutically, functional studies demonstrated gene therapy-delivered cardiac-specific MAPK7 restoration or overexpression of CEBPβ impeded cardiac injury after MI, at least partly due to normalization of *mir128-3p*. Furthermore, inhibition of *mir128-3p* preserved *Irs1* and ameliorated cardiac dysfunction post-MI. In conclusion, we reveal that targeting *mir128-3p* mitigates myocardial insulin resistance, thereafter slowing down the progression of heart failure post-ischemia.

## Introduction

Ischemic heart disease is a leading cause of mortality worldwide (*Benjamin et al., 2019*). Although significant advances in treatment have improved early survival after myocardial infarction (MI), such as bypass surgery, percutaneous coronary intervention, and thrombolytic therapy, the high prevalence of post-MI-associated heart failure (HF) remains (*Sanchis-Gomar et al., 2016*). Clinical studies suggest that promoting reliance on glucose under ischemic condition represents an indispensable adaption, which may have beneficial effects on survival in patients suffering from ischemic heart disease (*Montessuit and Lerch, 2013*; *Ritterhoff and Tian, 2017*). At an early stage post-ischemia, enhanced insulin-dependent glucose utilization in cardiomyocytes protects the heart against injury exacerbation (*Nagoshi et al., 2011*; *Ussher et al., 2012*). Conversely, myocardial insulin resistance precedes and prophesies the onset and progression of HF during prolonged stress (*Aroor et al., 2012*; *Ciccarelli et al., 2011*; *Fu et al., 2016*; *Riehle and Abel, 2016*). Impaired insulin signaling pathways in cardiomyocytes leads to loss of the major energy source, increased oxidative stress due

to metabolic shift toward fatty acids, and deleterious cardiac remodeling (*Aroor et al., 2012*; *Fu et al., 2016*; *Tian and Abel, 2001*). However, few studies have investigated the molecular pathogenesis of cardiac insulin resistance post-ischemia.

In the heart, insulin receptor substrate (IRS) is a key modulator in the insulin signal transduction pathway. Upon activation by insulin receptor (INSR), IRS facilitates glucose transporter 4 (SLC2A4, also known as as GLUT4) translocation, as well as initiates cell survival pathways in cardiomyocytes (*Guo and Guo, 2017*). Notably, human heart samples from subjects with advanced heart failure display an impaired IRS signaling and reduced plasma membrane SLC2A4 (*Chokshi et al., 2012*; *Cook et al., 2010*). Myocardial loss of both IRS1 and IRS2 renders the heart more vulnerable to cardiac dysfunction in mice suffering from diabetes (*Desrois et al., 2004*; *Qi et al., 2013*). In addition, cardiac IRS1 and IRS2 double knockout mice displayed contractile dysfunction in early life (*Riehle et al., 2013*). More importantly, impaired insulin sensitivity in ischemic myocardium is attributed to defective IRS1 function (*Ciccarelli et al., 2011*; *Fu et al., 2016*). Noting the protective role of cardiac IRS1, the feasibility of ameliorating cardiac insulin sensitivity and heart injury via maintaining IRS1 following ischemia has been underscored.

MicroRNAs (miRNAs) act as post-transcriptional gene regulators by either degrading target transcripts or suppressing their protein translation (*Ambros, 2004*). Signature alterations of cardiac miRNAs have been implicated in a multitude of pathological processes, including cardiac hypertrophy, ischemic injury and heart failure (*Caroli et al., 2013*; *Wojciechowska et al., 2017*). Therefore, silencing disease-prompted miRNAs using antagomiRs is considered as a potential therapeutic intervention for heart diseases (*van Rooij et al., 2008*). Approximately 50% of human miRNAs are expressed from the introns of encoding transcripts, referred to as intronic miRNAs (*Rodriguez et al., 2004*), which are transcribed from the primary transcripts (Pri-miRNAs) along with their host genes (*Saini et al., 2007*). *Mir128* is an intronic miRNA, which promotes cardiomyocyte apoptosis during ischemia/reperfusion injury in rabbits (*Zeng et al., 2016*), while its loss advances mouse cardiac regeneration after MI (*Huang et al., 2018*). However, whether *mir128* is implicated in myocardial insulin resistance post-MI is unexplored.

In this study, we seek to ascertain that cardiac IRS1 reduction is causative for HF progression post-MI via impairing insulin sensitivity in the myocardium. In addition, we demonstrate that upregulated *mir128-3p* directly targets the 3'-UTR of *Irs1* and represses its expression in response to hypoxic stress. *mir128-3p* is processed from *Pri-mir128-1* transcript, which is negatively regulated by the transcription factor CCAAT/enhancer-binding protein beta (CEBPβ). Phenotypic and molecular analyses revealed that CEBPβ suppression of *mir128-3p* and maintenance of IRS1-conferred insulin sensitivity under ischemic insult. Furthermore, this regulatory signaling cascade is reinforced by functional evidence showing that adeno-associated virus (AAV9)-delivered *Cebpb* (encoding CEBPβ) or *mir128-3p* antagomiR retain IRS1 expression and alleviate injury expansion in mouse hearts post-MI. Taken together, our findings unveil a novel cardio-protective mechanism that provides new insights into treatment strategies for heart failure by relieving insulin resistance in ischemic myocardium.

## Results

### IRS1 is downregulated in long-term ischemia-induced failing hearts

We first investigated whether cardiac dysfunction correlates with impaired glucose pathway following MI. To do so, we applied permanent occlusion of left anterior descending (LAD) coronary artery on male C57BL/6N mice. Four weeks post-MI, the mice displayed cardiac dysfunction exemplified by reduced fractional shortening (FS%), increased end-diastolic left ventricular internal diameter (dLVID) and end-systolic left ventricular internal diameter (sLVID) (*Figure 1A and B*). Deleterious heart function was coupled to severe infarction and interstitial fibrosis in the non-infarct region (*Figure 1C and D*). To assess genome-wide manifestation of glucose pathways, we performed RNA-Seq from the non-infarct area. We focused on analyzing the key genes involved in cardiac glucose metabolism, revealing that a significant number of genes responsible for glucose uptake and glycolysis were differentially expressed in the failing hearts post-MI (*Figure 1E*). Among these altered genes, IRS1 was prominently decreased, which was validated by quantitative PCR (qPCR) and immunoblots (*Figure 1F and G*); however, the levels of other factors involved in glucose uptake (*Insr*, *Slc2a4*, and *Slc2a1*) were not significantly changed (*Figure 1F and G*). As a result, the insulin

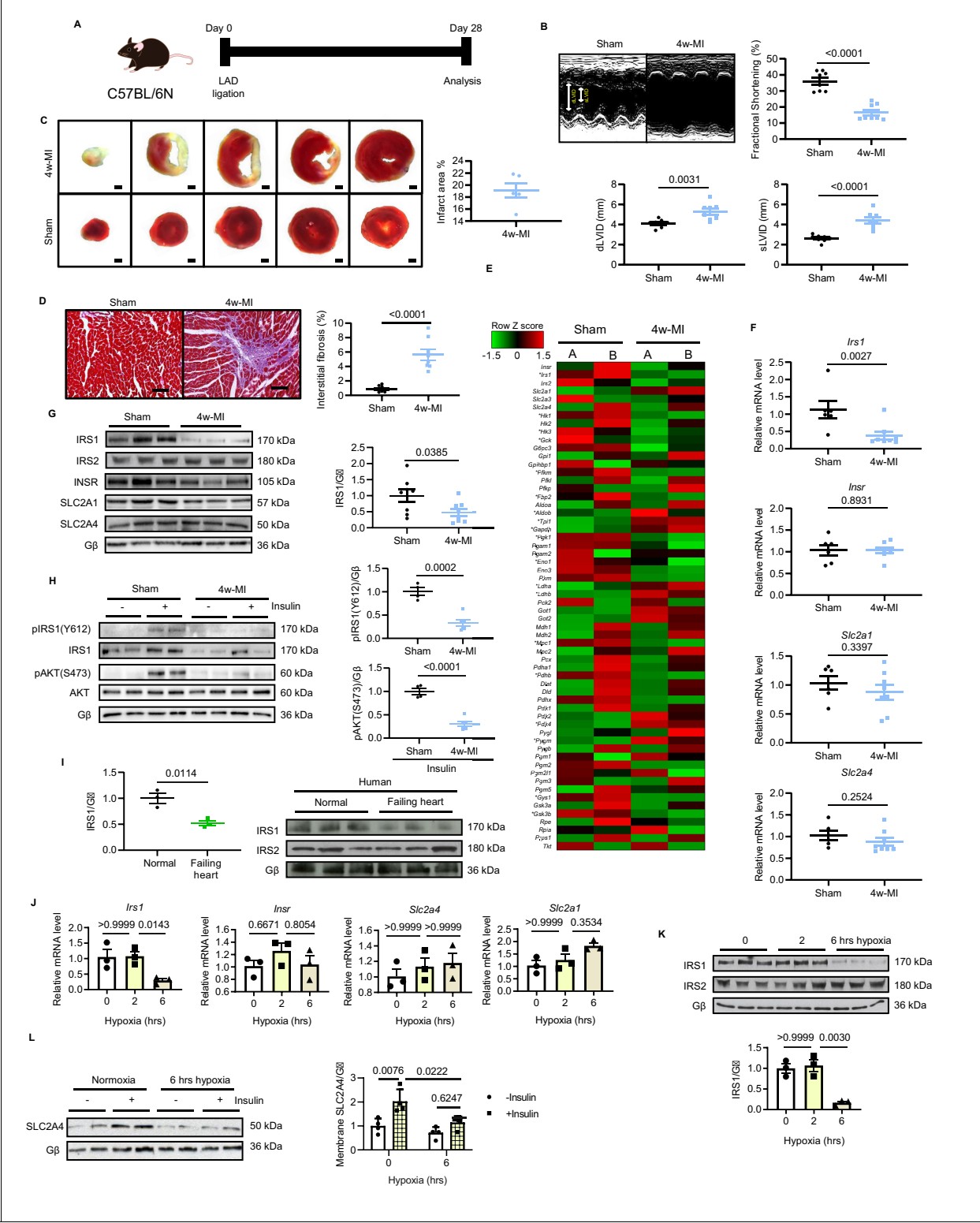

**Figure 1.** IRS1 is downregulated in failing hearts and prolonged hypoxia-stressed cardiomyocytes. (**A**) Experimental design of male C57BL/6N mice subject to 4w-MI. (**B**) M-mode images captured by echocardiography determined cardiac dysfunction after 4w-MI, measured by fractional shortening (FS%) and end-diastolic and systolic left ventricular internal diameter (dLVID and sLVID) (N = 8 mice). (**C**) Infarct area (from one heart) detected by triphenyltetrazolium chloride (TTC) staining (N = 5 mice). Scale bar: 1 mm. (**D**) Interstitial fibrosis in the non-infarct region determined by Masson's Trichrome staining (N = 8 mice). Scale bar: 100 μm. (**E**) RNA-sequencing (RNA-Seq) heat map depicts expression levels of the selected genes in the category of glycose pathways. * indicates genes identified as significantly different between sham and MI for four weeks (Log₂ fold change according to

*Figure 1 continued on next page*

*Figure 1 continued*

adjusted P value < 0.05) (N = 2 mice). **(F–G)** qPCR validation and immunoblot analyses of IRS1, IRS2, INSR, SLC2A1 (also known as GLUT1), and SLC2A4 (also known as GLUT4) in the myocardium (N = 6–8 mice). **(H)** Insulin-dependent pathway evaluated by immunoblots after insulin injection (1 U/kg for 30 mins). **(I)** IRS1 and IRS2 protein level in failing human hearts (N = 3 mice). **(J)** qPCR detection of *Insr, Irs1, Slc2a1,* and *Slc2a4* in ARCMs under hypoxia (N = 3 experiments). **(K)** IRS1, but not IRS2, was reduced in ARCMs under long-term hypoxia (N = 3 experiments). **(L)** Insulin stimulation-induced (0.1 µM, 30 mins) SLC2A4 membrane translocation in cardiomyocytes was diminished after 6 hr hypoxia (N = 4 experiments). Data presented as mean ± SEM. Comparisons between two groups were performed using two-tailed Student's t-test. One-way or two-way ANOVA followed by Bonferroni post hoc tests were employed as appropriate.

The online version of this article includes the following source data for figure 1:

**Source data 1.** Insulin sensitivity and *Irs1* downregulation in MI-induced heart failure and rat cardiomyocytes under prolonged hypoxic stress.

pathway was blocked, as indicated by diminished phosphorylation of IRS1 and AKT in response to insulin stimulation after 4 week MI (*Figure 1H*). More importantly, cardiac insulin resistance is a feature of human HF (*Aroor et al., 2012*; *Riehle and Abel, 2016*), and IRS1, but not IRS2, was found to be downregulated in failing human hearts (*Figure 1I*). In line with observations from mouse and human heart samples, IRS1 transcript level and protein expression were both decreased in adult rat cardiomyocytes (ARCMs) under prolonged hypoxic (1.5% O$_2$) stress (*Figure 1J and K*); however, the levels of *Insr, Slc2a4* and *Slc2a1* were unchanged (*Figure 1J*). SLC2A4 is the most abundant glucose transporter in adult cardiomyocytes residing in intracellular vesicles under basal condition. It translocates to the plasma membrane to facilitate insulin-dependent glucose entering cardiomyocytes in response to insulin (*Cook et al., 2010*). Intriguingly, insulin-stimulated plasma membrane-enriched SLC2A4 was significantly lower under long-term hypoxic condition (*Figure 1L*). These data suggest that downregulated IRS1 could be causative for impaired insulin-dependent glucose pathway in ischemic hearts.

## IRS1 is directly targeted by upregulated *mir128-3p*

We next asked how are IRS1 transcript and protein decreased in MI-induced failing myocardium. MiRNAs are able to suppress genes expression during heart failure progression (*van Rooij et al., 2008*). To address whether IRS1 is downregulated by any miRNA action, the 3'-untranslated region (3'UTR) sequence of *Irs1* was analyzed using TargetScan. Among three potential cardiac-abundant miRNAs (*mir128-3p, mir144-3p* and *mir145-5p*) targeting *Irs1* 3'UTR, only *mir128-3p* displayed upregulation in the myocardium of long-term MI-induced failing hearts (*Figure 2A*, *Figure 2—figure supplement 1A*). In situ hybridization of *mir128-3p* further confirmed an increased signal in ischemic myocardium (*Figure 2B*, *Figure 2—figure supplement 1B*). Consistently, increased *mir128-3p* was detected in ARCMs treated with prolonged hypoxia for 4–6 hr (*Figure 2C*, *Figure 2—figure supplement 2*). Mir128 is an intronic miRNA, which localizes in two gene *loci* in human, rat, and mouse: mir128-1 in *R3hdm1* and mir128-2 in *Arpp21* (*Figure 2D*, *Figure 2—figure supplement 3*). To investigate which of the gene *loci* is regulated under ischemic condition, we evaluated the transcripts of these two primary forms of mir128 (*Pri-miR-128–1* and *Pri-mir128-2*). Endogenous *Pri-mir128-1* level was augmented in sustained MI-induced failing hearts; however, no change of *Pri-mir128-2* was detected (*Figure 2—figure supplement 1C*). Consistently, *Pri-mir128-1* was increased in ARCMs stressed by 4 and 6 hr hypoxia (*Figure 2E*, *Figure 2—figure supplement 4*). Next, we evaluated whether *mir128-3p* inhibits *Irs1* expression. The seed sequence of *mir128-3p* is highly conserved among mouse, rat, and human (*Figure 2F*). In silico screening detected two potential binding sites in the 3'UTR of human *IRS1*, one of which is highly conserved across mammalian species (*Figure 2G*). *IRS1* 3'UTR luciferase reporter assay was performed to confirm *IRS1* as a direct target of *mir128-3p* in H9C2 cardiomyocytes. The validated *mir128-3p* mimic (*Figure 2—figure supplement 5A*) caused a decrease in luciferase activity with respect to the control plasmid; however, mutations that disrupt base pairing between *mir128-3p* and native *IRS1* 3'UTR upheld the luciferase expression (*Figure 2H*). More convincingly, enforced *mir128-3p* reduced both transcript and protein levels of IRS1 in ARCMs (*Figure 2I and J*). Conversely, *IRS1* 3'UTR luciferase activity was enhanced by eliminating hypoxia-induced endogenous *mir128-3p* using validated *mir128-3p* antagomir (*Figure 2—figure supplement 5B*, *Figure 2K*). As anticipated, antagomir maintained IRS1 expression in ARCMs albeit exposure of prolonged hypoxia (*Figure 2L and M*). Furthermore, we assessed whether manipulation of *mir128-3p* levels could affect glucose intake in cardiomyocytes. *Mir128-3p* mimic

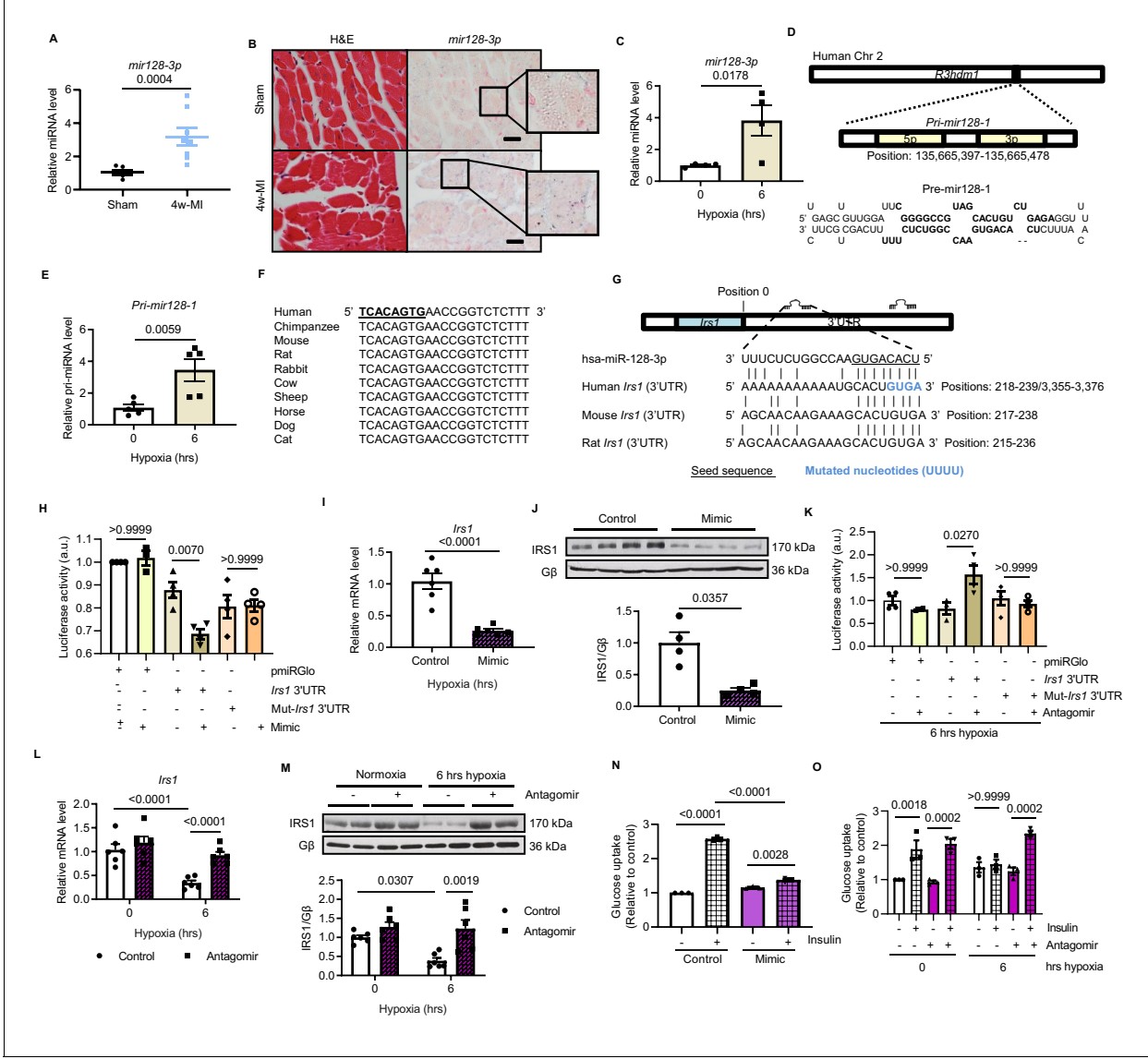

**Figure 2.** Hypoxia-upregulates *mir128-3p* and degrades *Irs1*. (**A**) Mature *mir128-3p* was increased in 4w-MI myocardium (N = 6–8 mice). Data were normalized against mature *mir191-5p* expression. (**B**) In situ hybridization with digoxigenin-labeled *mir128-3p* probes (right) was accompanied with consecutive section stained by H and E (left) (N = 3–5 mice). Scale bar: 20 μm. (**C**) Upregulation of *mir128-3p* in ARCMs after 6 hr hypoxia (N = 4 experiments). (**D**) Location of *Pri-mir128-1* hosted by human *R3HDM1* gene and stem-loop structure of *Pre-mir128-1*. (**E**) *Pri-mir128-1* level was determined by qPCR (N = 5 experiments). (**F**) Alignment of conserved *mir128-3p* sequence among mammals. (**G**) Sequences of *mir128-3p* conserved complementarity in human, mouse, and rat at *Irs1* 3'UTR. (**H**) *Irs1* 3'UTR luciferase reporter assay was performed in rat H9C2 cardiomyocytes transfected with pmiRGlo, pmiRGlo-*Irs1* 3'UTR, or mutant form (pmiRGlo-mut-*Irs1* 3'UTR). Luciferase activity was suppressed by *mir128-3p* mimic (N = 4 experiments). (**I–J**) qPCR and immunoblots illustrated miR-128–3 p mimic repression of IRS1 in ARCMs (N = 4–6 experiments). (**K**) *Irs1* 3'UTR luciferase reporter activity recovered by *mir128-3p* antagomiR (N = 4 experiments). (**L–M**) qPCR and immunoblots showed *mir128-3p*antagomiR maintained IRS1 expression in hypoxia-treated ARCMs (N = 4–6 experiments). Glucose uptake capacity by insulin stimulation (0.1 μM for 30 mins) was measured by the accumulation of 2-deoxy-D-glucose in ARCMs in the presence of (**N–O**) *mir128-3p* mimic or *mir128-3p* antagomiR (N = 3 experiments). Data presented as mean ± SEM. Comparisons between two groups were performed using two-tailed Student's t-test. One-way or two-way ANOVA followed by Bonferroni post hoc tests were employed as appropriate.

The online version of this article includes the following source data and figure supplement(s) for figure 2:

**Source data 1.** Study of *mir128-3p* expression under hypoxia and its function in regulating *Irs1*.

**Figure supplement 1.** MiRNA analysis in 4w-MI myocardium.

**Figure supplement 1—source data 1.** MiRNA analysis in 4w-MI myocardium.

**Figure supplement 2.** *Mir128-3p* level in rat cardiomyocytes after hypoxia.

**Figure supplement 2—source data 1.** *Mir128-3p* level in rat cardiomyocytes after hypoxia.

*Figure 2 continued on next page*

**Figure supplement 3.** *Pri-mir128-2* location and structure.
**Figure supplement 4.** *Pri-mir128-1* level in rat cardiomyocytes after hypoxia.
**Figure supplement 4—source data 1.** *Pri-mir128-1* level in rat cardiomyocytes after hypoxia.
**Figure supplement 5.** *Mir128-3p* overexpression and inhibition optimization.
**Figure supplement 5—source data 1.** *Mir128-3p* overexpression and inhibition optimization.

impeded insulin-stimulated glucose uptake (*Figure 2N*); on the contrary, antagomir showed retained glucose enter into ARCMs under hypoxia for 6 hr (*Figure 2O*). Taken together, these data reveal that sustained hypoxic circumstance triggers upregulation of *mir128-3p*, which eventually recognizes and binds to *IRS1* 3'UTR. Consequently, reduced IRS1 may cause impaired insulin-dependent glucose uptake in cardiomyocytes.

## CEBPβ represses the primary transcript of *mir128-3p* in hypoxia

Prompted by the observations that long-lasting hypoxic condition gave rise to increased *Pri-mir128-1* and mature *mir128-3p* levels, we speculated that this augmentation is regulated at the transcriptional level. Intronic miRNA can be transcribed along with its host gene (*Saini et al., 2007*). However, the promoter region localized 1.5 kbp downstream from transcription start site (TSS) of human *R3hdm1* exon1 did not respond to hypoxia (*Figure 3—figure supplement 1*). By further screening, *Pri-mir128-1* promoter luciferase reporter assay targeting another putative TSS located between exon1 and exon2 of *R3hdm1* illustrated that only long-term hypoxia (4 and 6 hr) promoted transcription of *Pri-mir128-1* in H9C2 cardiomyocytes (*Figure 3A*). Using Ensembl and UCSC Genome Browser, we found that there are CEBP consensus binding sites in this region, which are well conserved across species (*Figure 3B*). Interestingly, cardiac-abundant *Cebpb* (encoding CEBPβ), but not *Cebpa* (encoding CEBPα), was decreased in C57BL/6N mouse hearts subject to prolonged ischemia for four weeks (*Figure 3C*, *Figure 3—figure supplement 2A*). Similarly, *Cebpb* was also reduced by long-standing hypoxic stress in ARCMs (*Figure 3D*). CEBPβ is known as a critical co-transcription factor to activate or repress metabolic genes (*van der Krieken et al., 2015*). We hypothesized that CEBPβ (LAP isoform, the most dominant and functioning isoform of CEBPβ) acts to transcriptionally repress *mir128-3p*, which was supported by CEBPβ overexpression minimizing *Pri-mir128-1* and *mir128-3p* expression, and consequently retaining *Irs1* transcript despite prolonged hypoxia, (*Figure 3E*, *Figure 3—figure supplement 3C*). Conversely, rising *Pri-mir128-1* was detected in *Cebpb*-knockdown ARCMs in response to short-term hypoxia for 2 hr (*Figure 3—figure supplement 3D*, *Figure 3F*), which consolidated a negative correlation between *Cebpb* and *Pri-mir128-1* in hypoxic conditions. Subsequent to *Cebpb* deficiency, *Irs1* transcript was downregulated due to the increased mature *mir128-3p* (*Figure 3F*, *Figure 3—figure supplement 3D*). Additionally, *Pri-mir128-1* promoter luciferase activity was appreciably increased in H9C2 cardiomyocytes due to the loss of *Cebpb* (*Figure 3G*). Of note, *Cebpb* was not found to be related to *Pri-mir128-2* in cardiomyocytes (*Figure 3—figure supplement 3E*). As such, these data suggest that long-standing hypoxic condition downregulated CEBPβ, subsequently abolishing its repression on *mir128-3p* transcription.

## CEBPβ negative regulation of *mir128-3p* is mediated by MAPK7-CREB1 pathway

MAPK7 (also known as ERK5) is known to be cardio-protective by mediating transcription factor regulation of genes in response to pathological stresses (*Kimura et al., 2010*; *Liu et al., 2017*). Interestingly, we detected decreased MAPK7 in 4 week MI-induced failing hearts in C57BL/6N mice (*Figure 3H*, *Figure 3—figure supplement 2B*). Consistently, MAPK7 was also declined in ARCMs (*Figure 3I*) or neonatal rat cardiomyocytes (NRCMs) upon prolonged hypoxia for 6 hr (*Figure 3—figure supplement 4*). Of note, phosphorylation of MAPK7 was increased by short-term hypoxia, while abrogated by long-term stress (*Figure 3I*, *Figure 3—figure supplement 4*). It has been demonstrated that MAPK7 is related to the expression of IRS1 in the hearts under diabetic stress (*Liu et al., 2017*), we thereafter focused our attention on the plausible link of MAPK7 to CEBPβ and *mir128-3p* under hypoxic condition. Indeed, decreased *Cebpb* along with increased *mir128-3p* was shortly detected in *Mapk7*-deficient ARCMs after 2 hr hypoxia (*Figure 3J*); however, MAPK7 overexpression

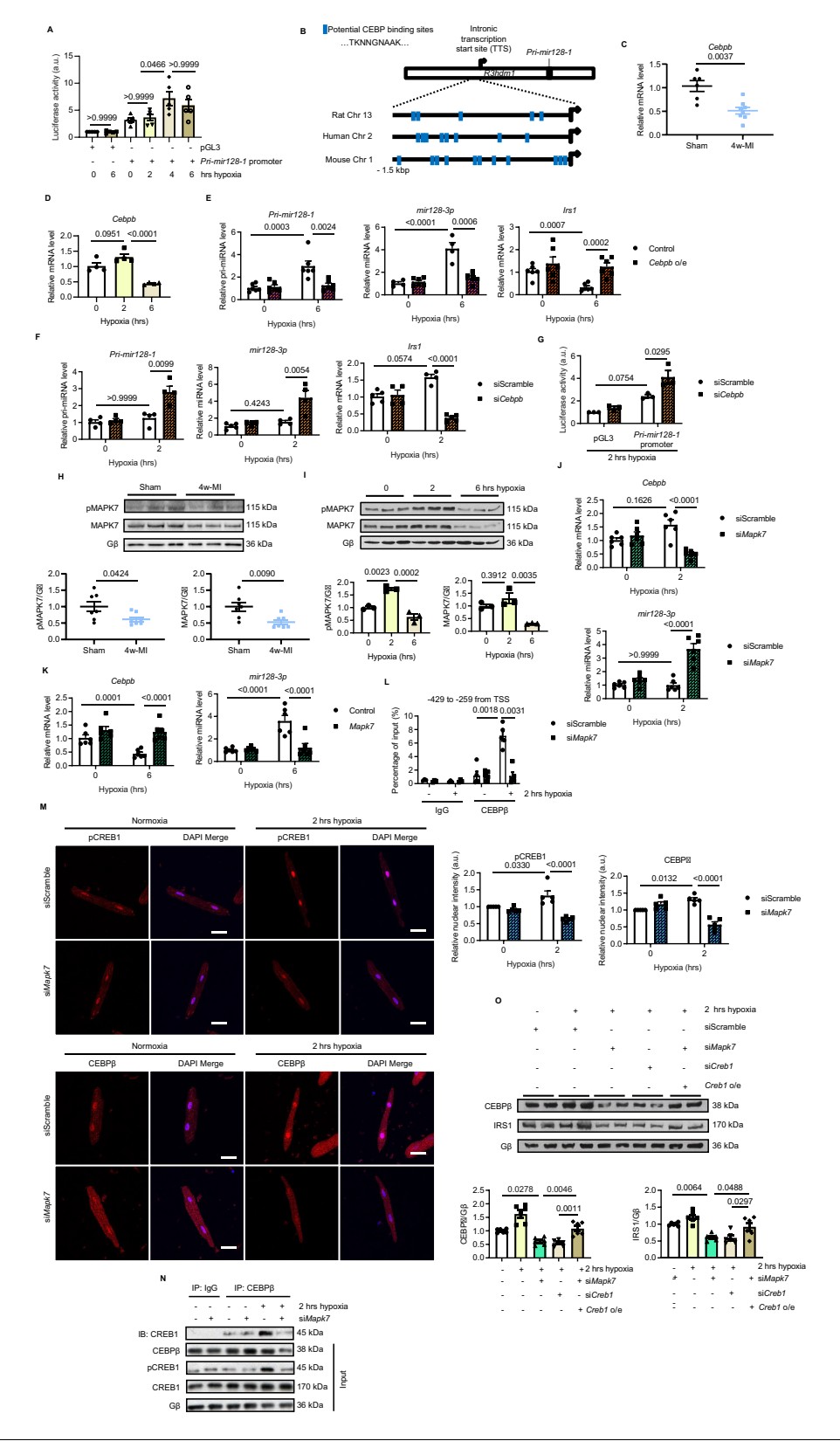

**Figure 3.** MAPK7 regulation of CEBPβ transcriptionally represses *mir128-3p*. (**A**) *Pri-mir128-1* promotor luciferase activity in H9C2 was increased by long-term hypoxia (N = 4 experiments). (**B**) Schematic figure showing CEBP consensus binding sites in the intronic promoter region between exon1 and

*Figure 3 continued on next page*

*Figure 3 continued*

exon2 of *R3hdm1* gene conserved in human, rat, and mouse. (C–D) *Cebpβ* was downregulated in myocardium after 4w-MI (N = 6–8 mice) and ARCMs (N = 4 experiments) stressed by short and long-lasting hypoxia. (E–F) Measurements of *Pri-mir128-1*, *mir128-3p* and *Irs1* were performed in ARCMs with CEBPβ overexpression (*Cebpb* o/e) under long-term hypoxia or *Cebpb* knockdown in response to short-term hypoxia (N = 4–6 experiments). (G) *Pri-mir128-1* promotor luciferase activity was augmented by CEBPβ deficiency (N = 3 experiments). (H) MAPK7 (also known as ERK5) phosphorylation and expression were decreased in the myocardium four weeks after MI (N = 7 mice). (I) MAPK7 was activated by acute hypoxic stress, while it was declined by prolonged stimulation in ARCMs (N = 3 experiments). (J–K) qPCR analyses of *Cebpb* and *mir128-3p* in *Mapk7*-deficient or *Mapk7*-overexpressing (*Mapk7* o/e) ARCMs (N = 6 experiments). (L) Chromatin Immunoprecipitation (ChIP) was performed on ARCMs DNA using anti-CEBPβ antibody (anti-IgG antibody as negative control) (N = 3–5 experiments). Hypoxia-induced CEBPβ binding to the proximal *Pri-mir128-1* promoter was inhibited by MAPK7 deficiency. Data were normalized to the input chromatin. (M) Immunofluorescent staining of ARCMs demonstrated that hypoxia-triggered nuclear translocation of phosphorylated CREB1 or CEBPβ (red) was blocked by *Mapk7* knockdown. DAPI was used for nuclear staining (blue) (N = 5 experiments). Scale bar: 30 μm. (N) Immunoprecipitation showed that hypoxia promoted association of phosphorylated CREB1 and CEBPβ in ARCMs; however, the affinity was reduced in the absence of MAPK7. IgG was used as negative control. (O) Decreased CEBPβ and reduced IRS1 were detected in the absence of MAPK7 or CREB1, while the reduction was rescued by CREB1 overexpression in *Mapk7*-knockdown ARCMs (N = 6 experiments). Data presented as mean ± SEM. Comparisons between two groups were performed using two-tailed Student's t-test. One-way or two-way ANOVA followed by Bonferroni post hoc tests were employed as appropriate.

The online version of this article includes the following source data and figure supplement(s) for figure 3:

**Source data 1.** Role of MAPK7 and CEBPβ in *mir128-3p* transcriptional regulation.
**Figure supplement 1.** Hypoxia effect on *R3hdm1* expression.
**Figure supplement 1—source data 1.** Hypoxia effect on *R3hdm1* expression.
**Figure supplement 2.** *Mapk7* and *Cebpa* expression in 4w-MI myocardium.
**Figure supplement 2—source data 1.** *Mapk7* and *Cebpa* expression in 4w-MI myocardium.
**Figure supplement 3.** Role of CEBPβ in *miR128-3p* and *Irs1* regulation in H9C2 cells.
**Figure supplement 3—source data 1.** Role of CEBPβ in *miR128-3p* and *Irs1* regulation in H9C2 cells.
**Figure supplement 4.** MAPK7 activation and expression in short and long-term hypoxia.
**Figure supplement 4—source data 1.** MAPK7 activation and expression in short and long-term hypoxia.
**Figure supplement 5.** CEBPβ binding analysis to *Pri-mir128-1* intronic promoter.
**Figure supplement 5—source data 1.** CEBPβ binding analysis to *Pri-mir128-1* intronic promoter.
**Figure supplement 6.** Analysis of MAPK7 downstream effectors after hypoxia.
**Figure supplement 6—source data 1.** Analysis of MAPK7 downstream effectors after hypoxia.
**Figure supplement 7.** Hypoxia-induced CREB1 and CEBPβ nuclear translocation in NRCMs.
**Figure supplement 7—source data 1.** Hypoxia-induced CREB1 and CEBPβ nuclear translocation in NRCMs.

preserved *Cebpb* and *mir128-3p* in a normal range even under long-term hypoxia for 6 hr (*Figure 3K*). The finding that MAPK7-mediated CEBPβ negatively regulates *mir128-3p* was further evidenced by chromatin immunoprecipitation (ChIP)-quantitative PCR (qPCR) assay. Short-term hypoxia promoted binding of CEBPβ to *Pri-mir128-1* promoter regions between exon1 and exon2 of *R3hdm1* containing consensus CEBP binding sequence; however, *Mapk7* knockdown led to a remarkable reduction of association at various putative binding regions (−429 to −259; −2550 to −2383; −4047 to −3873 and −4176 to −4026 from intronic TSS) (*Figure 3L*, *Figure 3—figure supplement 5*). These results indicate that MAPK7 activation is likely required for CEBPβ negative regulation of *mir128-3p* during short-term hypoxia; on the contrary, MAPK7 abrogation due to long-lasting hypoxia may lead to defective CEBPβ repression on *mir128-3p*.

Next, we made a further endeavor to explore how MAPK7 exerts its positive action on CEBPβ expression and activation. First, we screened potential MAPK7 downstream signaling factors responding to cardiac pathological stresses. Acute hypoxia-triggered CREB1 phosphorylation was inhibited by *Mapk7* knockdown in ARCMs (*Figure 3—figure supplement 6*). Moreover, immunofluorescent staining confirmed that short-term hypoxia-induced nuclei accumulation of activated CREB1 and CEBPβ; however, their signals were diminished as a result of *Mapk7* knockdown (*Figure 3M*). Consistent observations were gained in NRCMs (*Figure 3—figure supplement 7*), indicating the potential relevance of activated CREB1 and CEBPβ nuclear translocation. CREB1 is able to bind and positively facilitate CEBPβ expression and function (*Zhang et al., 2004*). Immunoprecipitation demonstrated that 2 hr hypoxia resulted in the physical association of phosphorylated CREB1 and CEBPβ; however, this affinity was unseen in the absence of MAPK7 (*Figure 3N*). To further evaluate whether CREB1, as a downstream effector of MAPK7, is also required for maintaining CEBPβ expression, we knocked down *Creb1* in rat cardiomyocytes, which displayed decreased CEBPβ and

reduced IRS1 expression following 2 hr hypoxia exposure (*Figure 3O*). The same effects detected from *Mapk7*-knockdown cells were rescued by CREB1 overexpression (*Figure 3O*). In summary, these data unveil that MAPK7 is required for activation of CREB1, thereafter upregulating CEBPβ as a transcriptional repressor of *mir128-3p* in response to short-term hypoxia. However, long-standing stress deprives the protective action of this signaling cascade.

## MAPK7 is a requisite for *mir128-3p* suppression and cardiac insulin sensitivity maintenance under ischemic condition

According to our observations that MAPK7 pathway was impaired in the myocardium after pro-longed ischemia/hypoxic stress, we aimed to investigate whether MAPK7 disruption contributes to cardiac detrimental function during ischemic stress. To obtain functional evidence regarding MAPK7 being required for CEBPβ negative regulation of *mir128-3p* to modulate cardiac insulin sensitivity following ischemic insult, we used cardiac *Mapk7*-deleted mice (*Mapk7*-cko) mice to imitate the pathological situation, in which MAPK7 loss occurred in the myocardium in wild-type mice with long-lasting ischemia (four weeks). In order to determine whether MAPK7 absence is a direct cause of cardiac insulin resistance, *Mapk7*-cko and their control littermates (*Mapk7^flox^*) were subject to short-term ischemia (one week) (*Figure 4A*). After one week, *Mapk7*-cko hearts promptly displayed detrimental function, determined by decreased FS% and enlarged left ventricular chamber dimensions; however, these impacts were not detected in *Mapk7^flox^* hearts, in which MAPK7 phosphorylation was triggered during short-term MI (*Figure 4B*, *Figure 4—source data 2*, *Figure 4—figure supplement 1A*). Additionally, even though cardiac Troponin I (cTnI) level was comparable in *Mapk7^flox^* and *Mapk7*-cko serum 24 hr after ligation, more elevated lactate dehydrogenase (LDH) concentration was observed in *Mapk7*-cko mice one week post-MI (*Figure 4C*), indicating MAPK7 loss led to aug-mented ligation-induced cardiac injury. Furthermore, cardiac MAPK7 disruption contributed to larger infarct area (*Figure 4D*) as well as more reactive myocardial fibrosis (*Figure 4E*). Hypertrophic growth in the ischemic hearts is a compensatory response after acute MI (*Olivetti et al., 2000*). Mor-phological analysis showed lower heart weight/body weight and cross-sectional area of cardiomyo-cytes in *Mapk7*-cko hearts compared to *Mapk7^flox^* hearts after MI (*Figure 4—figure supplement 1B and C*). Strikingly, accompanied by the enhanced phosphorylated MAPK7 detected in *Mapk7^flox^* hearts, CREB1 phosphorylation was also increased in response to 1 week MI; however, this activation was not detected in *Mapk7*-cko hearts (*Figure 4F*). Noticeably, in agreement with observations obtained from the failing hearts induced by 4 week MI, *Mapk7*-cko hearts exhibited abridged *Cebpb* expression, correlated with augmented *Pri-mir128-1* and *mir128-3p* compared to *Mapk7^flox^* hearts 1 week post-MI (*Figure 4G and H*, *Figure 4—figure supplement 2A*). The other two miRNAs poten-tially targeting *Irs1* (*mir144-3p* and *mir145-5p*) and *Cebpa* in the myocardium were not altered by MAPK7 ablation (*Figure 4—figure supplement 2B and C*). These data suggest that MAPK7 deple-tion leads to a more severe cardiac injury associated with defective CEBPβ repression on *mir128-3p* in response to ischemic stress.

We next attempted to examine the glucose uptake pathway in *Mapk7*-cko hearts under ischemia. Assessment of IRS1, INSR, SLC2A4 and SLC2A1 (also known as GLUT1 encoded by *Slc2a1*) expres-sion determined that only IRS1 was reduced in *Mapk7*-cko hearts after 1 week LAD ligation, whereas it was unchanged in *Mapk7^flox^* hearts (*Figure 4I and J*, *Figure 4—figure supplement 2D*). There-fore, the response to insulin was blunted in *Mapk7*-cko myocardium subject to 1 week MI (*Figure 4K*, *Figure 4—figure supplement 3A*). However, storage of glycogen was not significantly declined in *Mapk7*-cko hearts (*Figure 4—figure supplement 3B*). Consistent with in vivo results, IRS1 was reduced in *Mapk7*-knockdown ARCMs and NRCMs under short-term hypoxia, while no changes were detected in the control cardiomyocytes during this period (*Figure 4L and M*, *Fig-ure 4—figure supplement 4*). As expected, stimulated by insulin (0.1 µM for 40 mins), blocked acti-vation of IRS1 and AKT (*Figure 4N*) failed to recruit SLC2A4 onto the plasma membrane (*Figure 4O and P*) in the absence of MAPK7, which eventually resulted in reduced glucose uptake capability and ATP production despite short term hypoxia (*Figure 4Q and R*). Similar impaired glucose intake func-tion was observed when *Irs1* was knocked down (*Figure 4Q*). Finally, we demonstrated that defec-tive glucose uptake promoted cardiomyocytes death in face of hypoxia. For instance, profound cell death was detected upon loss of MAPK7 or IRS1 in a hypoxic environment, as evidenced by rising LDH released from ARCMs, as well as increased cleaved caspase three and the number of TUNEL positive nuclei (*Figure 4—figure supplement 5*). These results indicate that MAPK7 disruption

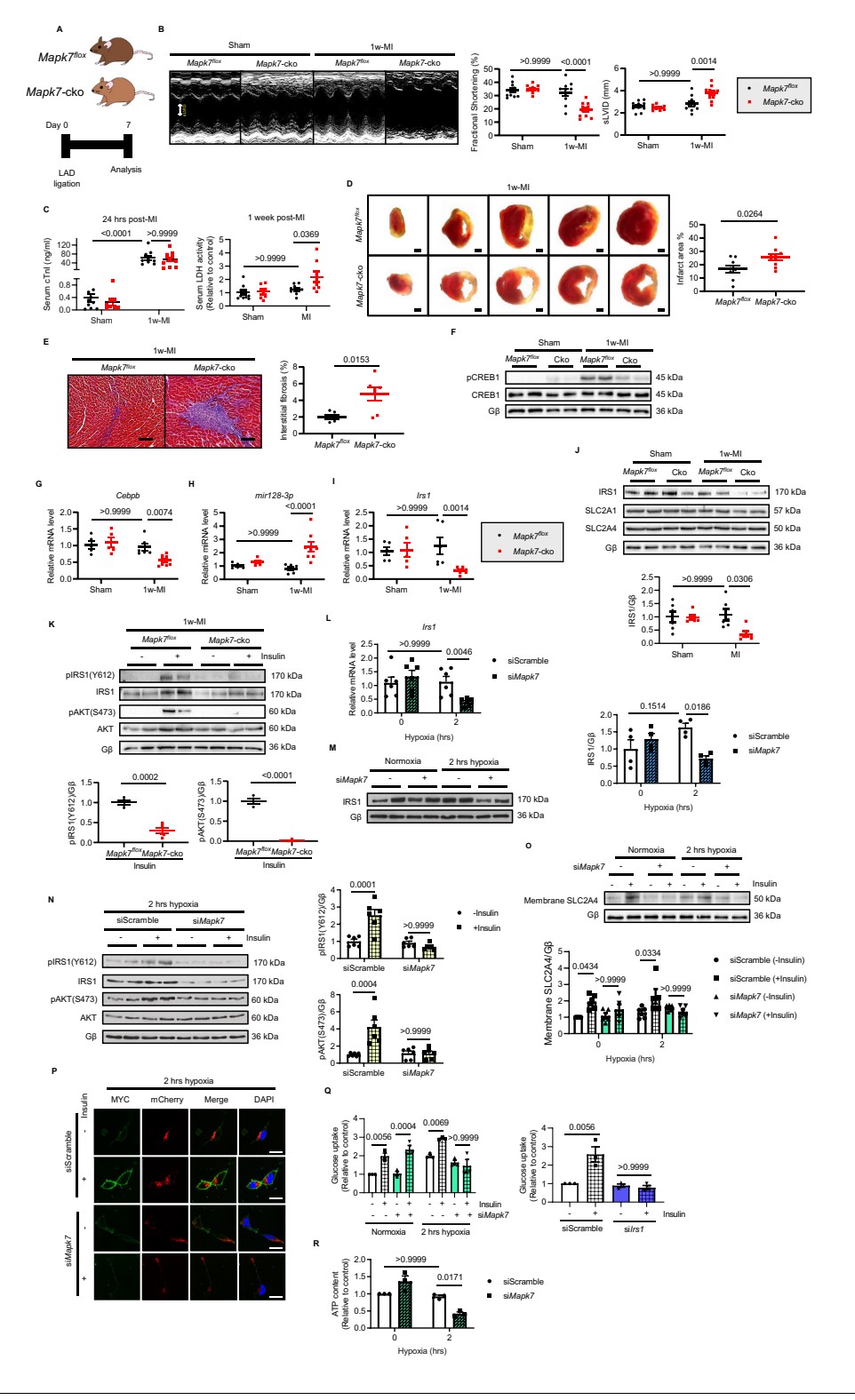

**Figure 4.** MAPK7 deficiency promotes cardiac insulin resistance and injury under hypoxic condition. (**A**) Experimental design of male cardiac-specific *Mapk7*-deleted mice subject to 1w-MI. (**B**) Echocardiography determined detrimental cardiac function in *Mapk7*-cko mice post-MI, indicated by fractional shortening (%) and sLVID (N = 9–11 mice). (**C–E**) MAPK7 ablation led to increased serum cTnI (24 hr post-ligation) and LDH (one week post-ligation) concentration, TTC-detected infarct area (from one heart, scale bar: 1 mm), and interstitial fibrosis in the non-infarct region detected by Masson's Trichrome (scale bar: 100 μm) (N = 8–10 mice). (**F**) Immunoblots analyses of phosphorylation and expression of CREB1 in the myocardium

*Figure 4 continued on next page*

*Figure 4 continued*

(N = 6 mice). (**G–H**) Real-time qPCR detected reduced *Cebpb* accompanied by upregulated mature *mir128-3p* due to *Mapk7* deletion in response to MI (N = 5–8 mice). (**I–J**) Transcript and protein level of IRS1 was decreased by MAPK7 disruption (N = 5–8 mice). (**K**) Myocardial insulin sensitivity was assessed by phosphorylation of IRS1 and AKT after insulin injection (1 U/kg for 30 mins) (N = 5–6 mice). (**L–N**) In ARCMs, the effects of *Mapk7* knockdown on *Irs1* transcript, IRS1 protein level, and insulin-stimulated (0.1 μM for 30 mins) phosphorylation of IRS1 and AKT were observed upon 2 hr hypoxia (N = 4–6 experiments). (**O–P**) SLC2A4 plasma membrane translocation was detected by immunoblots of the membrane fraction and confocal images of MYC-*Slc2a4*-mCherry immunofluorescence in H9C2 cells (N = 5 experiments). Fused mCherry (red) allows for SLC2A4 detection, while membrane translocated SLC2A4 is visualized by MYC (green). DAPI stains nuclear (blue). Scale bar: 25 μm. (**Q**) Insulin-dependent (0.1 μM for 30 mins) glucose uptake capacity under hypoxia was deferred in *Mapk7* or *Irs1* deficient ARCMs (N = 3 experiments). (**R**) ATP production was reduced in *Mapk7*-knockdown ARCMs under hypoxic condition (N = 3 experiments). Data presented as mean ± SEM. Comparisons between two groups were performed using two-tailed Student's t-test. Two-way ANOVA followed by Bonferroni post hoc tests was employed as appropriate.

The online version of this article includes the following source data and figure supplement(s) for figure 4:

**Source data 1.** Effect of MAPK7 deficiency on cardiac insulin resistance and in vivo and in vitro.
**Source data 2.** Echocardiographic parameters of *Mapk7*-cko and control mice one week post-MI.
**Figure supplement 1.** MAPK7 activation and hypertrophy in mouse myocardium one week post-MI.
**Figure supplement 1—source data 1.** MAPK7 activation and hypertrophy in mouse myocardium one week post-MI.
**Figure supplement 2.** Gene expression comparison between *Mapk7^flox* and *Mapk7*-cko mice one week post-MI.
**Figure supplement 2—source data 1.** Gene expression comparison between *Mapk7^flox* and *Mapk7*-cko mice one week post-MI.
**Figure supplement 3.** Activation of the insulin cascade in sham mice and cardiac glycogen detection one week post-MI.
**Figure supplement 3—source data 1.** Activation of the insulin cascade in sham mice and cardiac glycogen detection one week post-MI.
**Figure supplement 4.** Insulin-related gene expression in MAPK7-deficient rat cardiomyocytes under hypoxic stress.
**Figure supplement 4—source data 1.** Insulin-related gene expression in MAPK7-deficient rat cardiomyocytes under hypoxic stress.
**Figure supplement 5.** Effect of *Mapk7 and Irs1* knockdown in ARCMs cell death after hypoxia.
**Figure supplement 5—source data 1.** Effect of *Mapk7 and Irs1* knockdown in ARCMs cell death after hypoxia.

made the heart vulnerable to dysfunction through blockage of insulin-dependent pathways in the myocardium.

Given that *Mapk7*-cko mice exhibited impaired cardiac insulin sensitivity and cardiac function 1 week post-MI, we explored further evidence on whether MAPK7 restoration could slow down the progression of cardiac insulin resistance and heart failure in *Mapk7*-cko hearts stressed with ischemia. MAPK7 reconstitution in the myocardium was achieved by AAV9-delivery of a cardiac-specific troponin T (cTnT) promoter-driven full-length human *Mapk7* (AAV9-*Mapk7*, 1 × 10$^{11}$ genomic particles for injection, *Figure 5—figure supplement 1A*). AAV9-*Gfp* (as the control) or AAV9-*Mapk7* was injected into cardiac *Mapk7*-deficient mice, followed by 1 week LAD ligation (*Figure 5A*). MAPK7 restoration attenuated cardiac dysfunction (*Figure 5B*, *Figure 5—source data 2*). Meanwhile, although serum cTnI level 24 hr after ligation was not affected, LDH amount one week post-ligation suggested an alleviated cardiac injury by MAPK7 re-establishment (*Figure 5C*). Infarct area, interstitial fibrosis, and apoptosis rate in the non-infarct region were all reduced by AAV9-*Mapk7* administration (*Figure 5D and F*), while the hypertrophic response was recovered (*Figure 5—figure supplement 1B and C*). The restoration of MAPK7 to a physiological level reduced the infarction to a similar level observed in *Mapk7^flox* hearts (*Figure 5—figure supplement 2*). Notably, MAPK7 restoration rescued *Cebpb* level coupled to the minimized *mir128-3p* in *Mapk7*-knockout myocardium (*Figure 5G*). Next, we evaluated whether MAPK7 is able to protect the glucose uptake pathway in cardiomyocytes by retaining IRS1 expression. AAV9-*Mapk7* injection prevented IRS1 reduction in *Mapk7*-deleted hearts with 1 week ischemia (*Figure 5H and I*). Similar to the observation in heart tissues, *Irs1* transcript was maintained by MAPK7 overexpression in ARCMs in spite of long-term hypoxia (*Figure 5J*). As a consequence, glucose uptake was improved, which was determined by enhanced insulin-stimulated glucose intake rate in MAPK7-overexpressing cardiomyocytes (*Figure 5K*). Thus, MAPK7 is required for maintenance of IRS1 expression and retards the progression to cardiac dysfunction induced by ischemic insult.

## CEBPβ overexpression ameliorates MAPK7 loss-induced cardiac injury following MI

In a further attempt to obtain functional evidence supporting the proposed mechanisms responsible for MAPK7 regulation of CEBPβ signaling, we carried out rescue experiments on *Mapk7*-cko mice to

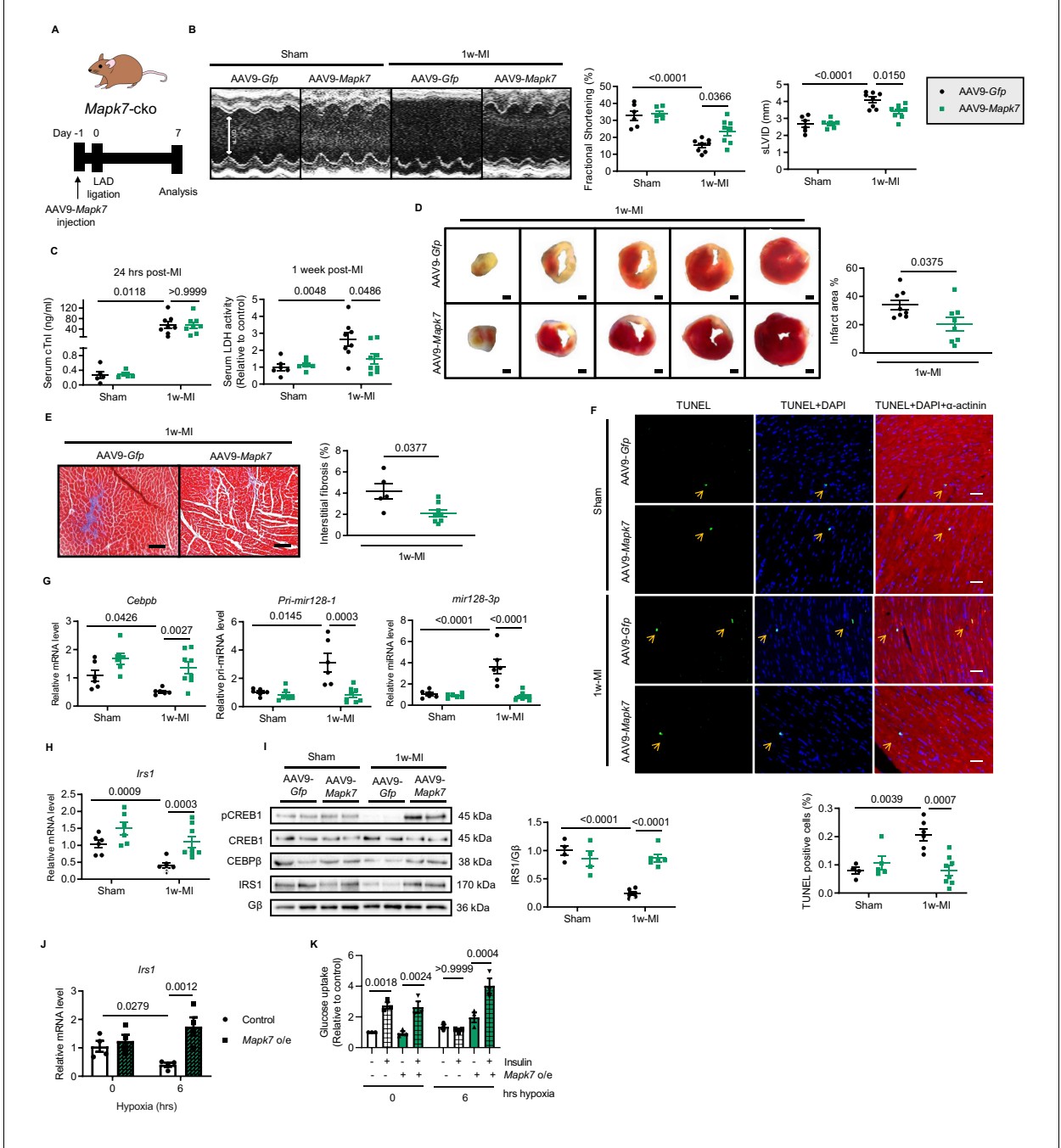

**Figure 5.** Cardiac MAPK7 restoration preserves IRS1 pathway in the myocardium and cardiac function in response to hypoxia. (**A**) Experimental design of cardiac MAPK7 restoration in *Mapk7*-cko mice via tail vein administration of AAV9-*Mapk7* ($1 \times 10^{11}$ viral particles), followed by LAD ligation for one week. (**B**) Echocardiography determined improved cardiac function by AAV9-*Mapk7* injection, measured by fractional shortening (%) and sLVID (N = 5–8 mice). (**C–F**) MAPK7 restoration decreased serum cTnI (24 hr post-ligation) and LDH (one week post-ligation), infarct area (scale bar: 1 mm), interstitial fibrosis in non-infarct area (scale bar: 100 µm), and apoptotic cells detected by TUNEL assay by triple staining (scale bar: 30 µm) of TUNEL (green), DAPI (blue for nuclear) and α-actinin (red for marking cardiomyocytes). Arrows indicate TUNEL positive nuclei (N = 4–8 mice). (**G**) qPCR showed that *Cebpb*, *Pri-mir128-1*, and mature *mir128-3p* were retained at the normal range by AAV9-*Mapk7* injection (N = 6–8 mice). (**H–I**) Transcript and protein level of IRS1 was maintained by MAPK7 restoration (N = 4–8 mice). (**J**) *Irs1* transcript in ARCMs was preserved by MAPK7 overexpression (*Mapk7* o/e) (N = 4 experiments). (**K**) Glucose uptake capacity of ARCMs was rescued by MAPK7 overexpression in response to insulin stimulation (0.1 µM for 30 mins) during long-term hypoxia (N = 3 experiments). Data presented as mean ± SEM. Comparisons between two groups were performed using two-tailed Student's t-test. Two-way ANOVA followed by Bonferroni post hoc tests was employed as appropriate.

The online version of this article includes the following source data and figure supplement(s) for figure 5:

*Figure 5 continued on next page*

Figure 5 continued

**Source data 1.** Functional and transcriptional effects of MAPK7 restoration in *Mapk7*-cko mice after MI and rat cardiomyocytes under hypoxia.
**Source data 2.** Echocardiographic parameters of *Mapk7*-cko mice injected with AAV9-Gfp or AAV9-Mapk7 one week post-MI.
**Figure supplement 1.** MAPK7 expression and cardiac hypertrophy in AAV9-*Mapk7*-injected mice one week post-MI.
**Figure supplement 1—source data 1.** MAPK7 expression and cardiac hypertrophy in AAV9-*Mapk7*-injected mice one week post-MI.
**Figure supplement 2.** Infarct area comparison between *Mapk7^flox^* and *Mapk7*-cko injected with AAV9-*Mapk7* seven days after MI.
**Figure supplement 2—source data 1.** Infarct area comparison between *Mapk7^flox^* and *Mapk7*-cko injected with AAV9-*Mapk7* seven days after MI.

evaluate whether overexpression of CEBPβ could prevent MAPK7 ablation-induced heart failure post-MI. We took advantage of AAV9-delivered cTnT promoter-driven *Cebpb* (AAV9-*Cebpb*, 1 × 10^11 genomic particles injection) to gain a 2-fold increase of cardiac CEBPβ expression (*Figure 6—figure supplement 1*). AAV9-*Gfp* (as the control) or AAV9-*Cebpb* was administrated in *Mapk7*-cko mice, followed by ischemic stress (*Figure 6A*). After one week, compared to AAV9-*Gfp* injected group, *Mapk7*-cko hearts receiving forced CEBPβ expression displayed improved cardiac performance (*Figure 6B*, *Figure 6—source data 2*). Myocardial injury and infarction rate were reduced in AAV9-*Cebpb* injected *Mapk7*-cko hearts (*Figure 6C and D*). Apparently, less fibrosis and cardiomyocyte death in the non-infarct region were also observed in *Cebpb*-overexpressed group (*Figure 6E and F*), albeit cardiac hypertrophy remaining akin to control group post-MI (*Figure 6G*). Importantly, the augmented *Pri-mir128-1* and mature *mir128-3p* in *Mapk7*-cko after 1 week ischemic insult was impeded upon CEBPβ overexpression (*Figure 6H*). As a consequence, IRS1 expression was retained by AAV9-*Cebpb* injection despite MAPK7 depletion (*Figure 6I and J*). Collectively, CEBPβ overexpression is able to mitigate cardiac injury via preventing *mir128-3p* suppression on IRS1.

## Silencing *mir128-3p* by AAV9-delivered antagomiR abrogates deleterious effects of cardiac insulin resistance post-MI

Likewise, we made a further effort to appraise whether prevention of upregulated *mir128-3p* is capable of maintaining Irs1 expression and cardiac insulin sensitivity, and consequently improve the contractile function. To do so, we constructed a long-term indigestible and RNA-polymerase III (H1)-driven tough decoy RNA (TuD) (*Xie et al., 2012*; *Xie et al., 2015*) under control of the cTnT promoter, termed *antimir128-3p*-TuD, to silence endogenous cardiac *mir128-3p* (*Figure 7A*). To validate the *antimir128-3p*-TuD activity, a luciferase reporter (pmiRGlo-*mir128-3p*) was constructed by adding four repeated *mir128-3p* complementary sequences to the 3' region of pmiRGlo plasmid. Most noticeably, *mir128-3p* mimic-suppressed luciferase activity was expectedly recovered by *antimir128-3p*-TuD (*Figure 7B*). As a result, IRS1, as a *mir128-3p* target, was preserved by *antimir128-3p*-TuD in cardiomyocytes (*Figure 7C*). Next, the validated *antimir128-3p*-TuD was transformed into an AAV9-based system (AAV9-*Antimir128*); therefore, we took advantage of AAV9-*Antimir128* on *Mapk7*-cko mice followed by MI for one week (*Figure 7D*). Compared to AAV9-Control miRNA injected groups, AAV9-*Antimir128* injection rescued cardiac dysfunction in MI-stressed *Mapk7*-cko mice (*Figure 7E*, *Figure 7—source data 2*). In addition to lower serum LDH after 1 week ligation, reduced infarct area, less fibrotic formation, and lower cell death rate were all observed in AAV9-*Antimir128*-injected *Mapk7*-cko myocardium post-MI (*Figure 7F and I*). However, cardiac hypertrophic growth was not affected by AAV9-mediated inhibition of *mir128-3p* (*Figure 7J*). Predictably, AAV9-*AntimiR128* application silenced the endogenous *mir128-3p* induced by MAPK7 disruption under ischemic insult, although without altering *Pri-mir128-1* transcript (*Figure 7K*). These outcomes were concomitant with maintained expression and activation of IRS1 and AKT (*Figure 7L and M*). Hence, the above data demonstrate that MAPK7 depletion rendered the mice unable to cope with ischemic stress, but inhibition of *mir128-3p* upregulation restored myocardial capacity to withstand it.

## Repression of *mir128-3p* by MAPK7-CEBPβ retains IRS1 in human iPSC-CMs

Finally, we sought human-relevant evidence for MAPK7-CEBPβ signaling pathway negative regulation of *mir128-3p* in response to hypoxia. To do so, human induced pluripotent stem cell-derived cardiomyocytes (iPSC-CMs) were treated with hypoxia (1.5% O₂) for various durations. First,

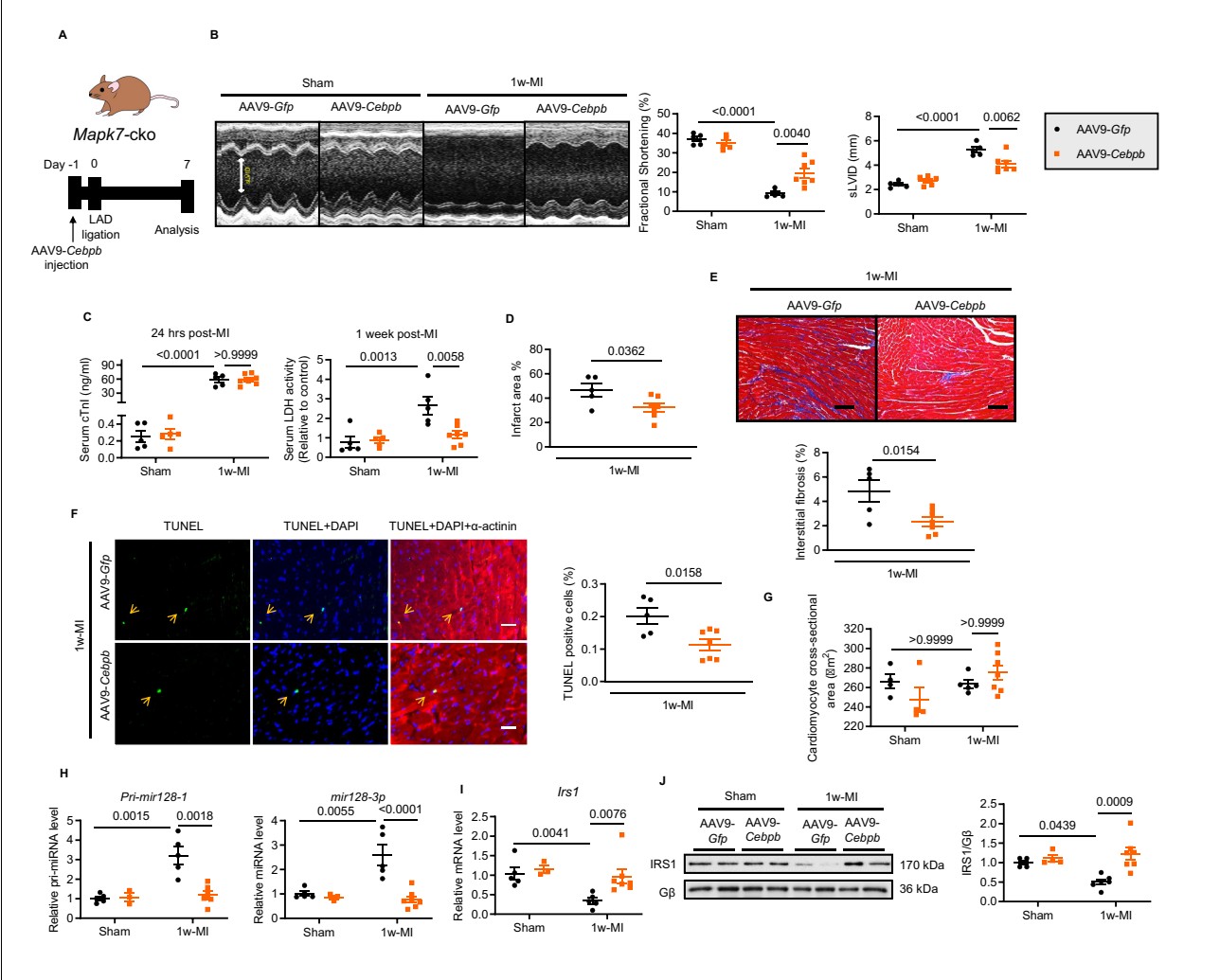

**Figure 6.** Cardiac-specific CEBPβ overexpression ameliorates cardiac injury in *Mapk7*-cko post-MI. (**A**) Experimental design of CEBPβ overexpression in *Mapk7*-cko myocardium via tail vein injection of AAV9-*Cebpb* (1 × 10^11 viral particles), followed by MI for one week. (**B**) Echocardiography showing preserved cardiac function by AAV9-*Cebpb* injection (N = 5–7 mice). (**C–F**) CEBPβ overexpression reduced serum cTnI (24 hr post-ligation) and LDH (one week post-ligation), infarct area, interstitial fibrosis in the non-infarct region (scale bar: 100 μm), and TUNEL positive apoptotic cells (green, scale bar: 30 μm), co-stained with DAPI (blue) and α-actinin (red). Arrows indicate TUNEL positive nuclei (N = 5–7 mice). (**G**) Cardiomyocyte size was measured from H and E stained sections (N = 4–7 mice). (**H**) *Pri-mir128-1* and mature *mir128-3p* levels were examined in the myocardium by qPCR (N = 3–7 mice). (**I–J**) qPCR and immunoblotting analyses showed the maintenance of IRS1 expression by CEBPβ overexpression after MI despite MAPK7 depletion (N = 3–7 mice). Data presented as mean ± SEM. Comparisons between two groups were performed using two-tailed Student's t-test. Two-way ANOVA followed by Bonferroni post hoc tests was employed as appropriate.

The online version of this article includes the following source data and figure supplement(s) for figure 6:

**Source data 1.** CEBPβ overexpression effect on *mir128-3p* and *Irs1 expression* in *Mapk7*-cko post-MI.

**Source data 2.** Echocardiographic parameters of *Mapk7*-cko mice injected with AAV9-Gfp or AAV9-*Cebpb* one week post-MI.

**Figure supplement 1.** Myocardial CEBPβ expression level after AAV9-*Cebpb* tail vein injection.

**Figure supplement 1—source data 1.** Myocardial CEBPβ expression level after AAV9-*Cebpb* tail vein injection.

decreased *Mapk7*, *Cebpb* and *Irs1* were obtained in iPSC-CMs after 6 hr hypoxia exposure (*Figure 8A*), which was consistent with the findings in mouse ischemic myocardium and rat cardio-myocytes (ARCMs and NRCMs) following long-lasting hypoxia. During short-term stress, increased MAPK7 activation was accompanied by elevated CREB1 phosphorylation and CEBPβ expression (*Figure 8B*). MAPK7 was a requisite for phosphorylation of CREB1, as made evident from the obser-vation that *Mapk7* knockdown (*Figure 8C*) prevented its nuclei accumulation induced by hypoxia in iPSC-CMs (*Figure 8D*). Subsequent to CREB1 activation, greater CEBPβ expression and activity was

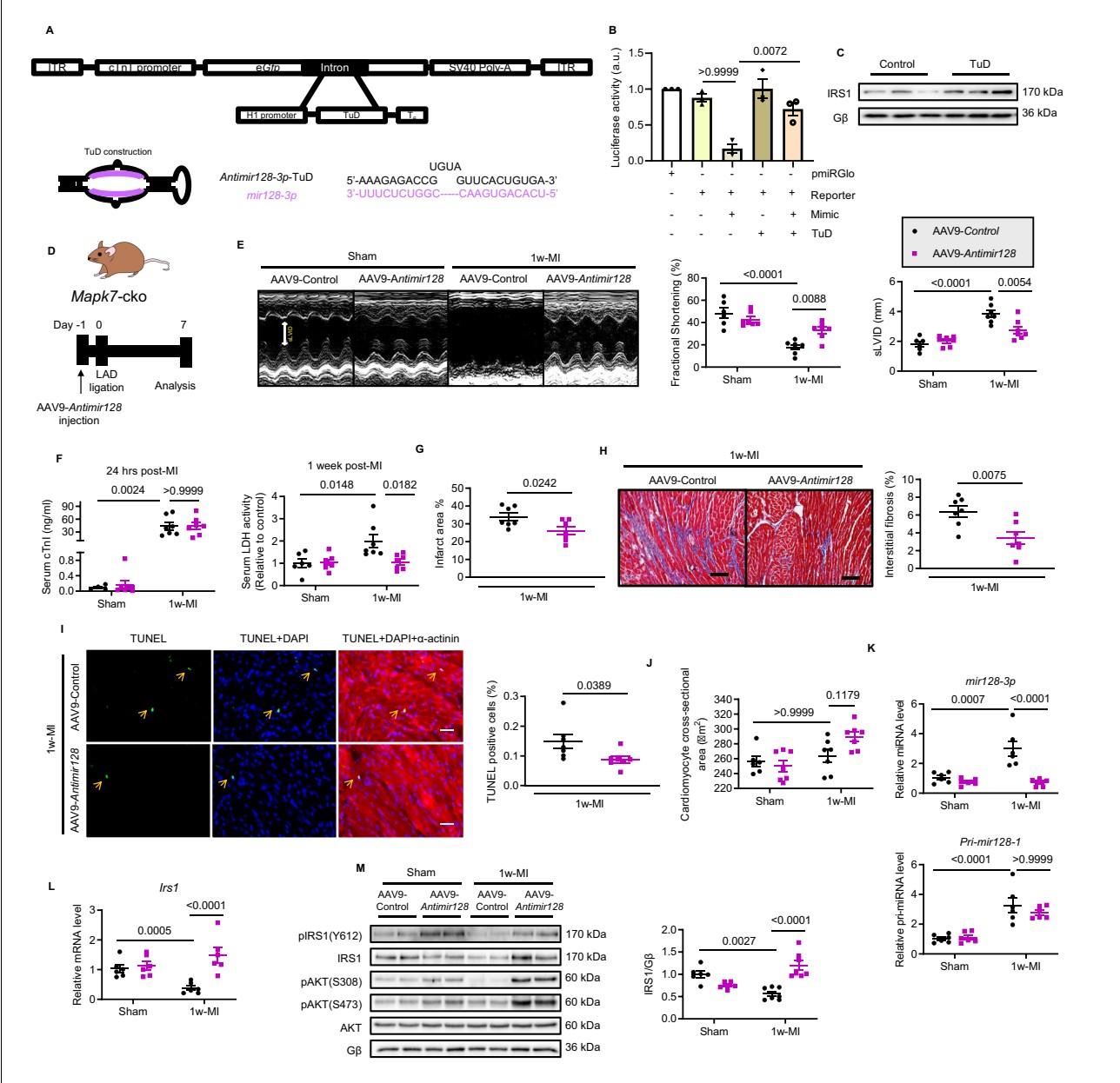

**Figure 7.** Silencing *mir128-3p* prevents *Irs1* degradation and decelerates HF progression post-MI. (**A**) Construction of antimir128-TuD built in the intronic region *Gfp* of AAV9-*Gfp* capsid. The sequence of antimir128-TuD is provided. (**B**) pmiRGlo-*mir128-3p* (four repeated complementary sequences of *mir128-3p* built in the 3' region of pmiRGlo) luciferase reporter activity was inhibited by *mir128-3p* mimic in H9C2 cardiomyocytes, but recovered by antimir128-TuD (N = 3 experiments). (**C**) Antimir128-TuD increased IRS1 protein level in cardiomyocytes (N = 3 experiments). (**D**) Experimental design of AAV9-delivered antimir128-TuD into *Mapk7*-cko mice by tail vein administration of AAV9-*antimir128* (1 × 10[11] viral particles), followed by MI. (**E**) Cardiac function was evaluated by echocardiography (N = 6–7 mice). (**F–I**) AAV9-*antimir128* inhibition of endogenous *mir128-3p* upregulated due to MAPK7 deficiency led to less serum cTnI (24 hr post-ligation) and LDH (one week post-ligation), infarct area, fibrosis (scale bar: 100 μm), and apoptosis is detected by TUNEL assay (green, scale bar: 30 μm), co-stained with DAPI (blue) and α-actinin (red). Arrows indicate TUNEL positive nuclei (N = 6–7 mice). (**J**) Cardiomyocyte size was measured according to H and E staining (N = 6–7 mice). (**K**) Mature *mir128-3p* was silenced by AAV9-antimir128 injection; however, *Pri-mir128-1* upregulation was not affected (N = 6 mice). (**L–M**) qPCR and immunoblotting analyses illustrated that expression and activation of IRS1 and AKT were preserved by AAV9-*antimir128* application in the absence of MAPK7 despite MI stress (N = 6–7 mice). Data presented as mean ± SEM. Comparisons between two groups were performed using two-tailed Student's t-test. One-way or two-way ANOVA followed by Bonferroni post hoc tests were employed as appropriate.

The online version of this article includes the following source data for figure 7:

**Source data 1.** Functional effect of silencing *mir128-3p* post-MI.
**Source data 2.** Echocardiographic parameters of *Mapk7*-cko mice injected with AAV9-Control or AAV9-*Antimir128* one week post-MI.

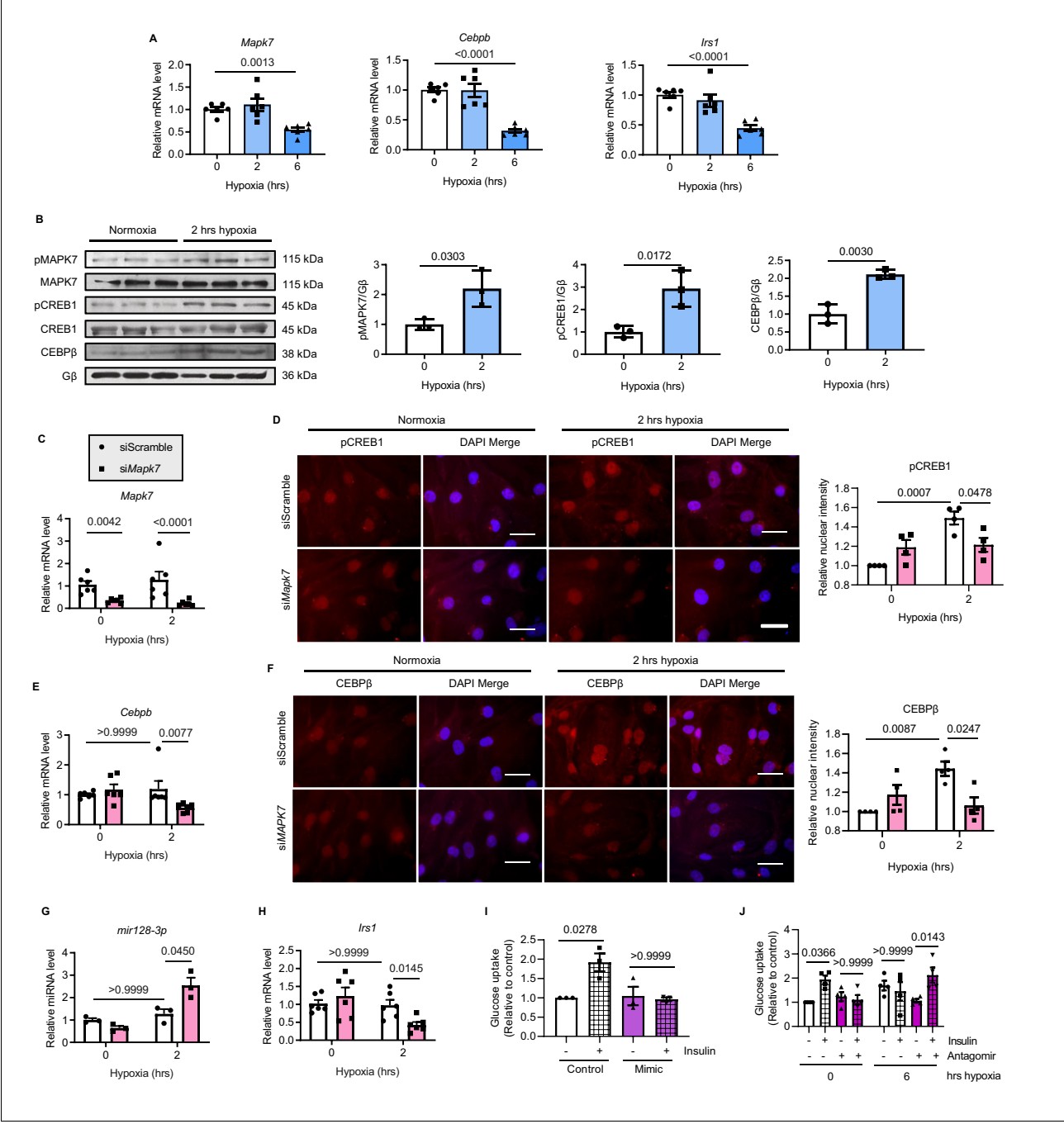

**Figure 8.** MAPK7-CEBPβ repression of *mir128-3p* maintains insulin sensitivity in human iPSC-CMs under hypoxia. (A) Transcripts of *Mapk7*, *Cebpb,* and *Irs1* were reduced in human iPSC-CMs in response to long-term hypoxia (N = 6 experiments). (B) Immunoblots and quantifications showed that phosphorylation of MAPK7 and CREB1 were accompanied by increased CEBPβ under short-term hypoxia (N = 3 experiments). (C) *Mapk7* knockdown using its siRNA in iPSC-CMs was validated by qPCR (N = 6 experiments). (D) Phosphorylated CREB1 nuclear accumulation (red) induced by short-term hypoxia. However, *Mapk7* knockdown blocked its nuclear deposition, counterstained with DAPI (blue) (N = 4 experiments, more than 50 nuclei measured per group per experiment). Scale bar: 20 μm. (E–F) qPCR and immunofluorescent staining (scale bar: 20 μm) determined that MAPK7 deficiency reduced expression and nuclear localization of CEBPβ (N = 4–6 experiments, more than 50 nuclei measured per group per experiment). (G–H) Augmented mature *mir128-3p* and decreased *Irs1* were detected by qPCR in *Mapk7*-knockdown iPSC-CMs under hypoxia (N = 3–6 experiments). (I–J) *mir128-3p* mimic abrogated insulin stimulation (0.1 μM for 30 mins)-induced glucose uptake in iPSC-CMs. On the contrary, *mir128-3p* antagomiR rescued insulin-dependent glucose uptake capability, albeit prolonged hypoxia circumstance (N = 3 experiments). Data presented as mean ± SEM. Comparisons between two groups were performed using two-tailed Student's t-test. One-way or two-way ANOVA followed by Bonferroni post hoc tests were employed as appropriate.

*Figure 8 continued on next page*

*Figure 8 continued*

The online version of this article includes the following source data for figure 8:

**Source data 1.** Analysis of the MAPK7-CEBPβ-*mir128-3p* axis in human iPSC-CMs under hypoxia.

determined by increased transcript level and its nuclear localization, whereas *Mapk7* deficiency markedly lessened CEBPβ expression and its nuclear accumulation (*Figure 8E and F*). Importantly, knockdown of *Mapk7* promoted mature *mir128-3p*, which was correlated with lower *Irs1* level in iPSC-CMs within short-term hypoxia stress (*Figure 8G and H*). Foremost, the insulin-dependent glucose uptake capability of iPSC-CMs was diminished by *mir128-3p* mimic (*Figure 8I*). In contrast, antagomir was able to protect glucose intake competency in iPSC-CMs under prolonged hypoxic stress (*Figure 8J*). Together, these data support that MAPK7-CEBPβ cascade preserves IRS1 expression through repression of *mir128-3p*, which has beneficial effects on the insulin-dependent pathway in human iPSC-CMs under hypoxic condition.

## Discussion

Myocardial insulin resistance is a cause for HF due to defective glucose utilization and insulin-dependent cell survival. Led by the detection of reduced IRS1 in the failing hearts, this study demonstrated: (1) impaired insulin signaling pathway in ischemic myocardium is a consequence of IRS1 degradation by *mir128-3p*, which at least partly results from defective CEBPβ repression on *Pri-mir128-1*; (2) molecular and functional evidence confirms that MAPK7 regulation of CEBPβ is required to maintain cardiac insulin sensitivity under ischemia; (3) AAV9-based overexpression of CEBPβ or inhibition of *mir128-3p* may preserve IRS1 pathway, therefore restraining cardiac injury expansion and delaying HF progression after MI, as illustrated in *Figure 9*. Our data provide new insights into the feasible therapeutic approach to deal with ischemic insult via correction of insulin resistance.

In the situation of hypoxia, glucose provides a small amount of ATP by glycolysis in the myocardium, which becomes a paramount compensatory energy source during short-term oxygen deprivation (*Montessuit and Lerch, 2013*; *Nagoshi et al., 2011*; *Ritterhoff and Tian, 2017*). RNA-seq data indicated distorted cardiac glucose pathways from non-infarct area of the failing hearts suffering from prolonged non-reperfused hypoxia, supporting the concept that cardiac insulin resistance contributes to the onset and development of heart failure in ischemic circumstances. Although there are concerns on whether the heart is able to endure the strain of obtaining energy predominantly from glycolysis for a long-term, sustained provision of glucose is particularly beneficial to the continual energy requirement of the heart during ischemia and can serve as a potential adjuvant treatment with current clinical interventions after MI.

Recent studies have described a causative relation between cardiac insulin resistance and cardiac dysfunction in pathological conditions, independent of diabetes mellitus (*Ciccarelli et al., 2011*; *Fu et al., 2016*; *Riehle and Abel, 2016*). In particular, cardiac insulin resistance manifests as insufficient IRS-facilitated glucose uptake and compromised AKT-involved cell protection in the myocardium (*Guo and Guo, 2017*). *Qi et al. (2013)* reported that mice with myocardial loss of both IRS1 and IRS2 die at the ages of 6–8 weeks due to HF under unstressed condition. It was also reported that cardiomyocyte-specific double knockout of IRS1 and IRS2 leads to cardiac dysfunction in early life by augmenting autophagy in neonatal heart (*Riehle et al., 2013*). Our study elicits that reduced IRS1 in the low-flow myocardial region contributed to accentuated cardiac injury, enlarged infarct area and impaired cardiac contractility, which could be the mechanistic explanation of HF onset and development during chronic stage post-MI. Consistently, we demonstrate that IRS1 deficiency in cardiomyocytes led to cytosolic retention of SLC2A4 and blunted ATP production, as well as intensified cell death. Of note, although IRS1 and IRS2 are homologous, they have distinct functions in various tissues. IRS2-mediated insulin pathway in liver and cancer cells is upregulated in hypoxic condition (*Mardilovich and Shaw, 2009*; *Wei et al., 2013*). However, IRS2 was not altered by ischemia/hypoxia in the cardiomyocytes, which even failed to compensate the harmful effects due to IRS1 reduction. The role and regulation of IRS2 in the myocardium will be further studied. Equally important, insulin-independent glucose uptake via SLC2A1 is considered as an essential compensatory factor for basal glucose entry. Several studies on mammals have detected upregulation of SLC2A1 at various durations of

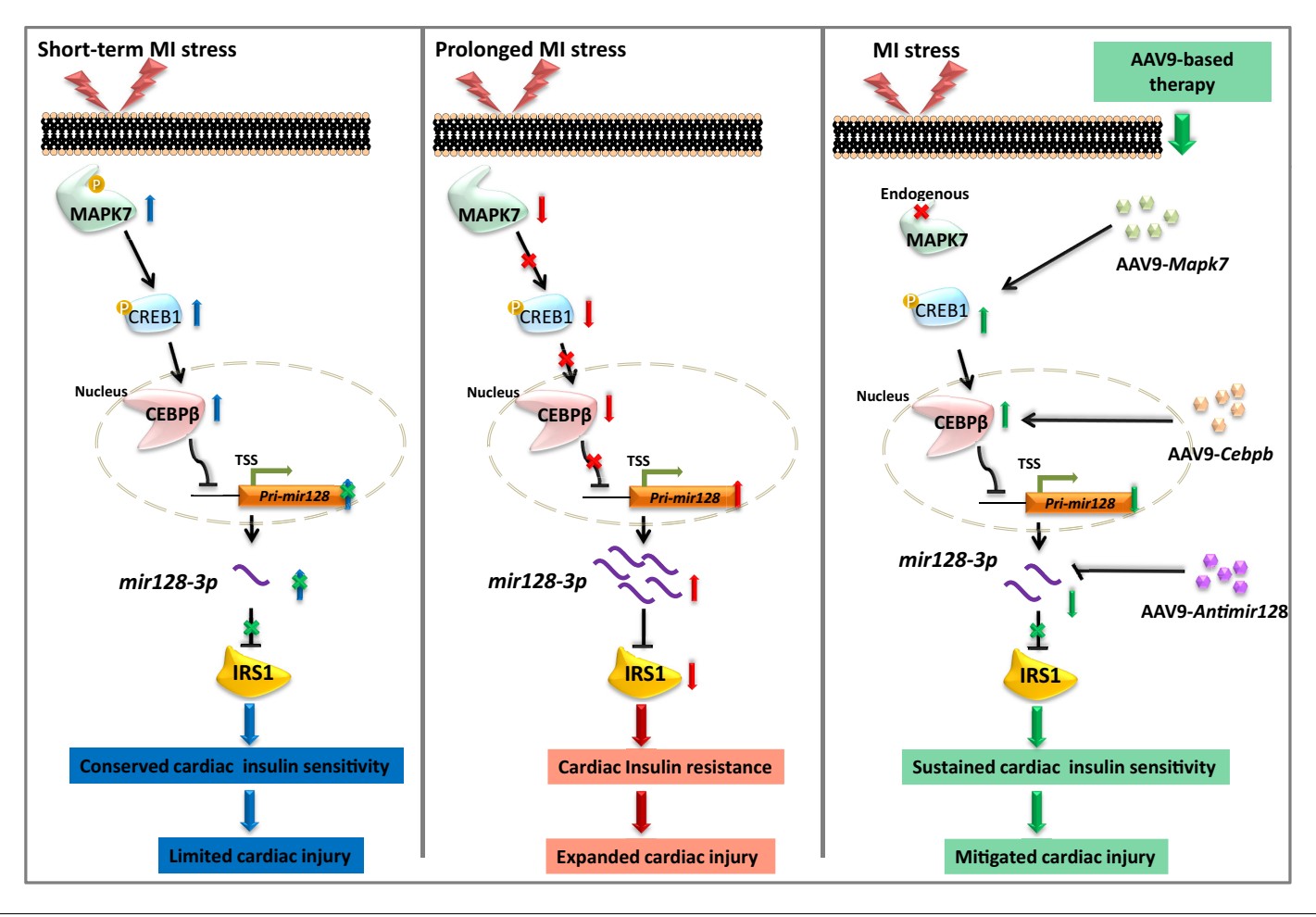

**Figure 9.** Schematic representation of MAPK7-CEBPβ signaling cascade negative regulation of *mir128-3p* under ischemia/hypoxic condition. Short-term ischemia/hypoxia activates MAPK7-CREB1, which facilitates CEBPβ expression and its nucleus translocation to act as transcriptional repressor on *Pri-mir128-1*, consequently maintaining IRS1 in the myocardium. However, prolonged stress induces MAPK7 loss and defective CEBPβ repression on *mir128-3p*, which results in IRS1 loss and impaired insulin pathway. Targeting this signaling cascade manifests the beneficial effects on maintenance of cardiac insulin sensitivity in response to ischemic/hypoxic stress, which is evidenced by AAV9-delivered cardiac MAPK7 restoration, cardiac-specific Cebpb overexpression, or silencing endogenous *mir128-3p*. List of source data files.

MI (*Brosius et al., 1997*; *Jóhannsson et al., 2001*; *Tardy-Cantalupi et al., 1999*). What is more, the hearts with SLC2A1 overexpression decrease aging-associated susceptibility to ischemic injury (*Luptak et al., 2007*). However, we did not detect a significant change of cardiac SLC2A1 in ischemia or hypoxia, indicating that the physiological level of SLC2A1 is insufficient to provide enough energy for maintenance of cardiac function under MI stress. These findings affirm the importance of strategies aiming to prevent cardiac insulin resistance as a potential therapy post-MI.

*Mir128*, dominantly expressed in neurons, negatively regulates stem cell-derived neurogenesis and neural system development (*Krichevsky et al., 2006*). In the heart, *mir128* inhibits postnatal cardiomyocytes proliferation by impeding cell cycle-related genes (*Huang et al., 2018*). In this report, insulin resistance was at least in part caused by augmented *mir128-3p* in the myocardium in response to ischemic stress. With regard to the reduction of Irs1 at transcript and protein levels, *mir128-3p* was identified to directly target *Irs1* 3'UTR (no *mir128-3p* binding site was found on *Irs2* 3'UTR) and stimulate its degradation. Eliminating *mir128-3p* was able to retain insulin-dependent glucose uptake despite facing hypoxic stress. Therefore, this study, for the first time, reveals that *mir128-3p* appears to play a role in cardiac glucose metabolism. In addition, since glucose utilization reinforces cardiomyocytes proliferation in early developmental stage (*de Carvalho et al., 2017*), our

findings may also provide another explanation for the previous observations that inhibition of *mir128-3p* enhances postnatal and adult cardiac regeneration (*Huang et al., 2018*).

Next, we have dissected the molecular basis underlying prolonged ischemia-upregulated *mir128-3p* in the heart. CEBPβ, primarily studied in liver and adipose tissue, modulates the transcription of a number of genes involved in metabolic regulation. Particularly, CEBPβ disruption results in dysregulation of genes facilitating glucose uptake (*van der Krieken et al., 2015*). In the heart, CEBPβ has been initially reported to repress exercise-induced cardiomyocyte growth and proliferation by negative regulation of serum response factor and ED-rich carboxy-terminal domain 4 (*Boström et al., 2010*). Contradictorily, CEBPβ is considered to promote phenylephrine-triggered cardiomyocyte growth via inhibition of nuclear factor kappa-light-chain-enhancer of activated B cells (*Zou et al., 2014*). Of note, there are three isoforms of CEBPβ (LAP*, LAP and LIP) differentially expressed in various tissues, being translated from a single mRNA (*van der Krieken et al., 2015*). LAP isoform was predominantly detected in the myocardium in this work, which is consistent with previous observations (*Boström et al., 2010*). A large number of studies describe CEBPβ (especially LAP isoform) as a transcriptional activator or repressor, including primary transcripts of miRNAs (*van der Krieken et al., 2015*). Multiple CEBP binding sites were found in promoter regions of *Pri-mir128-1*. Our study depicts that CEBPβ negatively regulates *Pri-mir128-1* in hypoxia, subsequently maintaining IRS1 expression. These findings corroborate that CEBPβ is involved in the regulation of cardiac metabolism.

We and others have revealed that cardiac MAPK7 (also known as ERK5) plays a protective role in response to pathological stresses (*Kimura et al., 2010*; *Le et al., 2012*; *Liu et al., 2017*; *Shishido et al., 2008*). Especially, MAPK7 facilitates the action of carboxyl terminus of Hsc7 interacting protein E3 ligase after MI in diabetic condition, which favors cell survival by preventing mitochondria-associated apoptotic pathways (*Le et al., 2012*). It is acknowledged that MAPK7 mediates insulin sensitivity in adipocytes (*Zhu et al., 2014*); however, there is a lack of conclusive evidence that MAPK7 is involved in myocardial insulin pathway regulation in the heart. This study once again highlights the cardioprotective role of MAPK7 in the ischemic myocardium. MAPK7 disruption resulted in a reduction of IRS1 in response to ischemia, akin to the previous observations in *Mapk7*-knockout diabetic hearts (*Liu et al., 2017*). As a consequence, distorted insulin pathway made the heart more vulnerable to cardiac dysfunction post-MI. In contrast, AAV9 gene-delivered MAPK7 restoration in *Mapk7*-cko mice maintained IRS1 expression and limited cardiac injury under ischemic insult.

The current study has shown MAPK7 phosphorylation was boosted after short-term MI, however, total MAPK7 expression declined in prolonged ischemia-stressed myocardium.Such an oscillation of phosphorylation and expression indicated that MAPK7 activation might manifest a biological function during acute hypoxia, while MAPK7 loss triggers the downstream signaling defects. Of note, *Shishido et al. (2008)* reported that MAPK7 phosphorylation is inhibited by ubiquitin-SUMOylation modification seven days after MI, particularly in type 1 diabetes. These discrepancies likely originate from the distinct backgrounds of experimental mice and the extent of ischemic injury. On the other hand, we have previously unveiled that MAPK7 expression is reduced in myocardium by protease-related degradation in high-fat-diet-induced diabetes mellitus (*Liu et al., 2017*). However, the mechanism whereby MAPK7 was downregulated in the ischemic myocardium after prolonged stress remains to be elucidated.

Our previous studies have revealed that MAPK7 functions through activation of MEF2 under hemodynamic or diabetic stresses (*Kimura et al., 2010*; *Liu et al., 2017*); however, the MEF2 family was not regulated by MAPK7 in hypoxia. CREB1, as another downstream effector of MAPK7, controls energy metabolism by adjusting key transcriptional regulators in various cell types (*Lin et al., 2005*). Our current findings agreed with the observations that CREB1 is activated in response to short-term ischemic/hypoxic stress to execute an anti-apoptotic function in cardiomyocytes (*Marais et al., 2008*). As such, cardiac MAPK7 governs distinctive signaling cascades in face of distinct pathological stresses. Activated CREB1 acts on CRE-like binding elements found in CEBPβ proximal promoter to induce its expression (*Zhang et al., 2004*). This study illustrates that upon activation, MAPK7 promotes the association of CREB1 and CEBPβ. It has been shown that CREB1 and CEBPβ interaction can lead to transactivation and transcriptional regulation in macrophages (*Ross et al., 2001*). However; how downregulated CEBPβ expression coincided with declined MAPK7-CREB1 under hypoxic condition will be further investigated.

Interestingly, CEBPβ expression was not altered by MAPK7 disruption in normoxia; moreover, its overexpression did not affect heart function at basal level. These observations suggest that CEBPβ

may be dispensable for IRS1-mediated pathways in unstressed situations. However, the functional evidence that deleterious effects in *Mapk7*-cko mice post-MI were salvaged by CEBPβ overexpression affirms that CEBPβ cascade is responsible for maintenance of cardiac insulin sensitivity under ischemic condition. Noticeably, *Huang et al. (2012)* have delineated that CEBPβ mediates reactivation of embryonic epicardial gene program and that intra-pericardial injection of adenovirus expressing dysfunctional CEBPβ confers resistance to ischemia-reperfusion injury. Therefore, whether CEBPβ is a feasible target for treating heart disease needs further in-depth examination.

It is well established that ischemia provokes cardiac hypertrophy, including compensatory hypertrophy and transitioned hypertrophy to heart failure depending on duration and extent of stress. We demonstrate that smaller cardiomyocyte size was observed in the non-infarct area of *Mapk7*-cko hearts compared to *Mapk7^flox^* hearts subject to LAD ligation for one week. Similarly, cardiac-specific *Irs1/2* double knockout mice elicit thinning of the cardiac muscle walls associated with more cell death (*Qi et al., 2013*), which is assumed to be a feature of failed compensatory hypertrophy. Hence, these observations expand the mechanistic explanation for *Mapk7*-deficient hearts displaying more vulnerability to HF progression in response to pathological stresses.

The myocardium has no ATP reserves, instead, exhibits a capability of storing fuel in the form of glycogen and triacylglycerol (*Ritterhoff and Tian, 2017*). Human cardiomyopathy is partially attributable to an imbalance of glycogen synthesis and utilization upon exposure to pathological stresses (*Adeva-Andany et al., 2016*). Glucose derived from glycogenolysis may provide energy supply for cardiac contraction in a low-flow condition (*Schaefer and Ramasamy, 1997*). RNA-seq detected decreased GYS1 and GSK3B in ischemic myocardium after prolonged ischemia, which may imply an additional mechanism exacerbating ischemic injury. In addition, MAPK7 disruption led to a trend of lower glycogen content in myocardium post-MI. This effect is plausibly caused by over-usage of stored glycogen due to MAPK7 loss-induced defective glucose supply, as well as faulty insulin-mediated glycogen synthesis. However, whether MAPK7 is involved in the regulation of glycogen synthesis and utilization in the myocardium requires further investigation.

Foremost, based on the consistent results gained from mature human iPSC-CMs under hypoxia, the human-relevant evidence of MAPK7-mediated double negative signaling regulation on IRS1 provided new insights into the therapeutic options for cardiac insulin resistance. Interestingly, it has been well accepted that silencing miRNAs have therapeutic potential in clinical settings. Using a long-lasting cardiac tough decoy RNA, we demonstrate that inhibition of upregulated endogenous *mir128-3p* counteracted ischemic injuries. These observations are in line with a recent study that *mir128* inhibition attenuates cardiomyocytes death and is beneficial for cardiac repair after ischemia/reperfusion (*Huang et al., 2018*). The evidence that *mir128-3p* directly degrades *Irs1* significantly advances our knowledge on molecular regulation of insulin signaling pathway in ischemic myocardium. Notably, IRS1 is dysregulated in hearts and other tissues in diabetes (*Lavin et al., 2016*). As such, this study hypothetically suggests that targeting *mir128-3p* has therapeutic potential for insulin resistance in or beyond the heart, as prevalently observed in diabetes mellitus.

In summary, we have uncovered a protective role of MAPK7-CEBPβ signaling cascade under ischemic insult by negative regulation of *mir128-3p* to maintain cardiac IRS1-dependent insulin sensitivity. Long-term ischemia-induced MAPK7 deficiency may account for defective CEBPβ repression on *mir128-3p*, consequently leading to *Irs1* degradation. Either preserving MAPK7-CEBPβ or directly impinging *mir128-3p* appears to be a potential approach to mitigate cardiac injury and could be an adjunctive treatment option in combination with surgical interventions for patients with ischemic heart disease.

## Materials and methods

A list of key resources can be found in *Supplementary file 1* – Key resources table.

### Study approval

All animal studies were performed in accordance with the United Kingdom Animals (Scientific Procedures) Act 1986 under the Home Office licence P3A97F3D1, and were approved by the University of Manchester Ethics Committee.

## Animals models

All mice and rats were housed in a pathogen-free facility at the University of Manchester. C57BL/6N mice and Sprague-Dawley rats were purchased from Envigo (UK). *Mapk7-floxed* (*Mapk7$^{flox}$*) mice were generated previously (*Ananieva et al., 2008*) with LoxP elements flanking exon4. Cardiomyocyte-specific *Mapk7* knockout mice (*Mapk7*-cko) were bred by mating *Mapk7$^{flox}$* mice with mice expressing Cre under the a-myosin heavy chain promoter (αMHC) (*Liu et al., 2017*). Mice were backcrossed into C57BL/6J background for at least six generations. Wild type littermates (*Mapk7$^{flox}$*) were used as controls. Mice with the same genotype were assigned to two or four groups by randomizing software (www.randomizer.org). All in vivo studies were blinded for genotype, treatment, and surgical procedure during the measurement and analysis stages. The only criteria for exclusion were technical failure or sudden death.

## Human heart samples

Human myocardial protein extracts from heart failure patients suffering from post-ischemic and dilated cardiomyopathy, and normal subjects were purchased from Asterand (US lab, Hertfordshire, UK). Information of case ID and sample ID were obtained by Asterand. The use of human tissue samples was approved by the United Kingdom Human Tissue Authority.

## Coronary artery ligation and insulin injection

Myocardial infarction was induced by permanent ligation of the left coronary artery. Only male mice aged around 12 weeks old were used. Mice were anesthetized by 3% isofluorane inhalation followed by intubation. The heart was exposed by left-side thoracotomy and the left coronary artery was ligated above its bifurcation by left atrium appendage. Ischemia was confirmed by the appearance of a pale area towards the apex. For sham surgery, the suture was pulled all the way through, but not tied. Buprenorphine (0.1 mg/kg) was administered subcutaneously at the beginning of the surgery, and the dose was repeated after 24 hr. For the study of cardiac insulin response, some mice were administered 1 U/kg of human insulin via IP injection 30 mins before collection of the heart.

## Cardiac troponin measurement

Infarction was verified by measurement of the cardiac troponin I (cTnI) in serum using the Ultra-Sensitive Mouse Cardiac Troponin-I Elisa (CTNI-1-US, Life Diagnostics). Blood was collected 24 hr post-MI from tail vein, left for 1 hr at room temperature to clot, and centrifuged for 15 mins at 6000xg to obtain the serum.

## Echocardiography

For cardiac function evaluation, mice were anesthetized with 1.5% isofluorane (Isothesia, Henry Schein). M-mode images were obtained by ultrasound imaging (Vevo 770, VisualSonics) to measure left ventricle chamber and wall dimensions. Fractional shortening was calculated for each mouse.

## AAV9 production and administration

Adeno-associated virus (AAV9)-based delivery system was used to overexpress MAPK7 and CEBPβ in vivo. Human *Mapk7* (a gift from Cathy Tournier's lab, University of Manchester) and human *Cebpb* (isoform of LAP, ID #15738, Addgene) cDNA were cloned into pSSV9-TnT-*Gfp* plasmid. For AAV9 production, each genome plasmid was co-transfected into low passage HEK293T cells along with the adenoviral helper plasmid (pDGΔVP) and the pAAV2-9 Rep-Cap plasmid (p5E18-VD2/9). Recombinant AAV9 was purified by discontinuous iodixanol gradient ultracentrifugation and titrated by qPCR (*Werfel et al., 2014*). A dose of $1 \times 10^{11}$ viral particles was administered to each mouse by intravenous tail vein injection. AAV9-*Gfp* was injected as a control.

## *Antimir128* tough decoy (TuD) design

To inhibit endogenous *mir128-3p*, a hairpin decoy structure was designed and delivered by AAV9 system. The *mir128-3p* sequence was obtained from miRBase (*Kozomara et al., 2019*). Two complementary sequences were placed on opposite sides of a long stem-loop structure, in accordance with the optimized structure of a tough decoy (*Haraguchi et al., 2009*; *Xie et al., 2012*). *Antimir128*-TUD was synthesized by Sigma. The forward and complimentary strands were annealed and cloned

into the intronic *Gfp* region of pSSV9-TnT-H1-inteGFP plasmid employing SalI and BamHI restriction sites. GFP expression is triggered by a cardiac troponin promoter, while TuD expression is initiated by RNA polymerase III targeting H1 promoter. As a negative control, the complementary strand for *C. elegans mir39-5p*, that does not target mammalian sequences (*D'Souza et al., 2017*), was incorporated into a TuD structure. AAV9 virus containing pSSV9-TnT-H1-*antimir128*-inteGFP (AAV9-*Antimir128*) or pSSV9-TnT-H1-control-inteGFP (AAV9-Control) were produced as described above.

- *Antimir128*-3p sequence: CAGTCGACGGGACGGCGCTAGGATCATCAACAAAGAGACCGG TGTATTCACTGTGACAAGTATTCTGGTCACAGAATACAACAAAGAGACCGGTGTATTCACTG TGACAAGATGATCCTAGCGCCGTCTTTTTTCCGGATCCGG
- Negative control sequence: CAGTCGACGGGACGGCGCTAGGATCATCAACAAGGCAAGC TGACCCTGAAGTTCAAGTATTCTGGTCACAGAATACAACAAGGCAAGCTGACCCTGAAG TTCAAGATGATCCTAGCGCCGTCTTTTTTCCGGATCCGG

## Determination of infarct size

For the measurement of the infarct area, metabolically active tissue was stained with triphenyltetrazolium chloride (TTC). After frozen on dry ice for 2 hr, the harvested hearts were cut into 1 mm thick sections perpendicular to the long axis, as shown below. The sections were incubated in 1% w/v TTC (T8877, Sigma) prepared in 0.1 mol/L phosphate buffer pH 7.4 for 15 mins at 37° C. Following, the heart sections were fixed in 4% PFA for 15 mins before imaging with a Leica M205 FA upright Stereomicroscope. The images were analyzed using ImageJ. Infarct size was calculated by measuring white areas (infarct region) against the total area.

## LDH assay

LDH activity in serum samples and cell supernatant was measured by colorimetric assay (MAK066, Sigma). Serum was prepared by allowing blood to clot for 1 hr at room temperature and centrifuging for 15 mins at 6000xg. The supernatant was measured directly. The assay was performed following the manufacturer's instructions. Briefly, samples were set up on a 96-well-plate, and 50 µl of Master Reaction Mix were added per well. The plate was shaken for 1 min and then incubated at 37° C. Absorbance at 450 nm was read every minute for 30 mins.

## Histology

Immediately after dissection, the hearts were fixed in 4% PFA for 4 hr at 4° C. They were subsequently washed with PBS for 30 mins and dehydrated in 50%, 70%, 90% and 100% ethanol for 2 hr each. The hearts were finally cleared in xylene overnight followed by paraffin infiltration. Blocks were trimmed and sectioned into 5 µm sections.

## In situ hybridization

Endogenous *mir128-3p* in the myocardium was detected with the miRCURY LNA miRNA Detection Probes (339111, Qiagen) on 6 µm paraffin sections. Sample preparation and hybridization were performed using miRCURY LNA miRNA ISH Buffer Set (339450, Qiagen). Briefly, the tissue was permeabilized with proteinase K for subsequent incubation with the double DIG-labelled LNA probe. Following stringent washing and blocking, samples were incubated with sheep anti-DIG AP FAB fragments (11093274910, Sigma). Freshly prepared AP substrate (11697471001, Roche) was applied for 2 hr at 30° C, which was followed by nuclear counterstaining with Nuclear Fast Red solution (N3020, Sigma). Consecutive sections were stained with H and E for morphological comparison. Brightfield images were captured using an Axiovision upright microscope equipped with a Coolsnap ES camera (Photometrics).

## Masson's Trichrome

Fibrotic tissue was differentiated by Masson's Trichrome staining. Heart sections were dewaxed, rehydrated, and fixed for 2 hr in Bouin's Solution (HT10132, Sigma). This was followed by nuclear staining with Harry's Hematoxylin (LAMB/230, RA Lamb Dry Chemical Stains) for five mins. Subsequently, Red Solution (HT151, Sigma) and Aniline Blue (B8563, Sigma) incubations were performed to stain muscle and collagen fibers, respectively. Finally, sections were dehydrated in ethanol and xylene baths, and mounted with the xylene-based medium, Eukitt (03989, Sigma). Brightfield images

were captured using an Axiovision upright microscope equipped with a Coolsnap ES camera (Photometrics) or D-Histech Panoramic-250 microscope slide-scanner. For quantification, ten fields were acquired per sample, and images were analyzed using ImageJ.

## Hematoxylin and Eosin (H & E) Staining

Cardiomyocyte cross-sectional area was calculated from H & E stained myocardial paraffin sections. After dewaxing and rehydration, sections were incubated in Harry's Haematoxylin (LAMB/230, RA Lamb Dry Chemical Stains) for five mins, followed by differentiation with 1% HCl prepared in 70% (v/v) ethanol for 10 secs. Blueing was achieved by washing with warm running tap water, followed by Eosin (6766007, Thermo Scientific) counterstain for 1 min. Samples were dehydrated and mounted with Eukitt. Brightfield images were captured using an Axiovision upright microscope equipped with a Coolsnap ES camera (Photometrics) or D-Histech Panoramic-250 microscope slide-scanner. For quantification, ten fields were acquired per sample, and images were analyzed using ImageJ. At least 250 cardiomyocytes were measured per heart.

## Periodic acid staining

Glycogen was detected using the Periodic Acid-Schiff (PAS) Staining System (395B, Sigma Aldrich) following the manufacturer's microwave procedure. PAS can stain glycogen, glycoproteins and glycolipids; therefore, relative glycogen content was calculated from the difference in staining intensity between consecutive sections that were stained with and without diastase treatment. Diastase treatment degrades glycogen, allowing for the differentiation of glycogen-specific staining. Brightfield images were captured using an Axiovision upright microscope equipped with a Coolsnap ES camera (Photometrics). For quantification, ten fields were acquired per sample, and images were analyzed using ImageJ.

## TUNEL

Apoptosis was detected in heart sections and ARCMs with triple staining: DAPI, TUNEL (In Situ Cell Death Detection Kit, Roche), and actinin (A7811, Sigma). TUNEL was performed following the manufacturer's instructions, followed by overnight incubation with primary α-actinin antibody at 4° C. Next, α-actinin signals were obtained by incubation with donkey anti-mouse-Alexa-594 conjugate (715-585-150, Jackson ImmunoResearch) for 2 hr at room temperature. Finally, the sections were mounted with DAPI-containing Vectashield (H-1200, Vector Laboratories). Fluorescent images were acquired with an Olympus BX51 upright microscope and a Coolsnap EZ camera (Photometrics). Specific band pass filter sets for DAPI, FITC and Texas red were used. For quantification, ten fields were acquired per sample, and images were analyzed using ImageJ.

## Adult cardiomyocytes (ARCMs) Isolation

Six-week-old male Sprague Dawley rats were administered a lethal dose of pentobarbital (150 mg/kg) with 250 U of heparin. The heart was removed and cannulated for retrograde perfusion through the aorta. The heart was washed for 5mins with perfusion buffer ($Ca^{2+}$ free Hank's buffered salt solution, 5.6 mmol/L glucose, 1 mmol/L $MgSO_4$) followed by addition of collagenase (117 U/ml, collagenase type 2, Worthington) and protease (0.175 U/ml protease type XIV, Sigma). After 20 mins of digestion, the left ventricle was cut into pieces and further digested in perfusion buffer containing collagenase and protease, in addition to 0.02 g/l trypsin, 0.02 g/l DNAse I, 1 mmol/L $CaCl_2$. Digested and filtered ARCMs were resuspended in ACCT mediums (DMEM+Glutamax medium (21885, Gibco) containing 1% BSA, 5 mmol/L L-carnitine, 2 mmol/L creatine, 5 mmol/L taurine, and 10 μmol/L blebbistatin). Finally, ARCMs were plated out and cultured in Geltrex (A1413302, Gibco) coated plates. Fresh ACCT medium was replaced every day before treatment.

## Human iPSC-derived cardiomyocytes (iPSC-CMs)

Human induced pluripotent stem cells (iPSCs) were derived from peripheral blood cells and were maintained in feeder-free culture conditions with E8 medium (A1517001, Gibco) on Geltrex-coated plates (A1413302, Gibco) as previously described (*Streckfuss-Bömeke et al., 2013*). Standard directed cardiomyocyte differentiation of iPSCs was initiated at a confluence of 90–100% via Wnt signaling modulation using cardiac differentiation medium (RPMI1640 HEPES Glutamax, 500 mg/L

human recombinant albumin, 200 mg/L L-ascorbic acid 2-phosphate). Progressive treatment was applied with 4 µmol/L CHIR99021 (361559, Millipore) for 48 hr and 5 µmol/L IWP2 (681671, Millipore) for further 48 hr. Medium was changed to cardiac culture medium (RPMI1640 HEPES Glutamax, B27 with insulin) at day 10. Cardiomyocytes were enriched using cardiac selection medium (RPMI 1640 minus Glucose, 4 mmol/L Lactate, 500 mg/L human recombinant albumin, 200 mg/L L-ascorbic acid 2-phosphate) for 5 days. iPSC-CMs were then cultured in cardiac culture medium up to 120 days for maturation before any further treatment.

## Cell hypoxia treatment and insulin stimulation

To mimic the hypoxic condition following a myocardial infarction, culture plates were introduced into a humid chamber and flushed with nitrogen to reduce oxygen concentration to 1.5% as measured by the fiber-optic oxygen sensor (FOSPOR-R, Ocean Optics). The sealed chamber was placed inside a cell culture incubator and kept at 37° C until the experimental endpoint was reached according to individual experiments in Results. Indicated cell groups were stimulated with 0.1 µmol/L insulin for 30 mins before collection.

## Cell transfection

H9C2 cells were authenticated by the ATCC and tested negative for mycoplasma regularly. Gene knockdown or overexpression in these was achieved with Lipofectamine 2000 Reagent (11668–019, Invitrogen), while Lipofectamine LTX and Plus Reagent (15338–100, Invitrogen) was used on ARCMs and iPSC-CM. Cells were transfected with 100 nmol/L of control siRNA (AGGUAGUGUAAUCGCC UUG, Sigma), rat *Mapk7* siRNA (AAAGGGUGCGAGCCUAUAU, Sigma), rat *Creb1* siRNA (Silencer Select s135439, Ambion), rat *Cebpb* siRNA (Silencer Select s127566, Ambion), rat Irs1 siRNA (Silencer Select s129870, Ambion), human *Mapk7* siRNA (SignalSilencer 7301, Cell Signaling), human *Mapk7* cDNA (a gift from Cathy Tournier's lab), human *Creb1* cDNA (ID #82203, Addgene), or human *Cebpb* cDNA (isoform of LAP, ID #15738, Addgene) following the manufacturer's instructions. Cells were incubated for 48 hr at 37° C with 5% $CO_2$ before further analysis.

For *mir128-3p* inhibition or induction in ARCMs and iPSC-CMs, Lipofectamine RNAiMAX Reagent (13778–075, Invitrogen) was utilized. Cells were transfected with 10 nmol/L of *mir128-3p* mimic (PremiR miRNA precursor PM114746, Ambion) or 40 nmol/L of antagomir (AntimiR miRNA inhibitor AM11746, Ambion) following the manufacturer's instructions.

## Subcellular fractionation

To separate cellular membrane and cytosolic fraction for detection of SLC2A4 translocation, H9C2 cells were lysed with Tris buffer (20 mmol/L Tris, 100 mmol/L NaCl, 2 mmol/L EDTA, 25 mmol/L glycerophosphate, 1 mmol/L $Na_3VO_4$, 1 mmol/L phenylmethanesulfonylfluoride, 1.54 µmol/L aprotinin, 21.6 µmol/L leupeptin, pH 7.6). Homogenates were centrifuged at 3000xg for 10 mins. The supernatant was centrifuged again at 200,000xg for 1 hr. The resulting supernatant was designated the cytosolic fraction, and the pellet was resuspended in RIPA buffer (150 mmol/L NaCl, 50 mmol/L tris, 0.1% w/v SDS, 0.25% w/v sodium deoxycholate, 2 mmol/L EDTA, 5% v/v glycerol, 1% v/v Triton X-100, 25 mmol/L glycerophosphate, 1 mmol/L $Na_3VO_4$, 1 mmol/L phenylmethanesulfonylfluoride, 1.54 µmol/L aprotinin, 21.6 µmol/L leupeptin, pH7.4) to obtain the membrane fraction. Protein concentration was quantified by Bradford assay (500–0006, Bio-Rad). Immunoblotting was performed with SLC2A4 (sc-7938, Santa Cruz Biotechnology) antibody.

## Immunoblotting

Total protein lysates from non-infarcted ventricular tissue (1–3 mm adjacent to infarct) or cells were prepared with Triton lysis buffer (137 mmol/L NaCl, 20 mmol/L Tris, 0.1% w/v SDS, 2 mmol/L EDTA, 10% v/v glycerol, 1% Triton X-100, 25 mmol/L glycerophosphate, 1 mmol/L $Na_3VO_4$, 1 mmol/L phenylmethanesulfonylfluoride, 1.54 µmol/L aprotinin, 21.6 µmol/L leupeptin, pH7.4). Lysates were cleared by centrifuging for 30 mins at 14,000xg. Protein concentration was quantified by Bradford assay (500–0006, Bio-Rad). Immunoblot analysis was performed with 30 µg of protein lysate using the following primary antibodies: pMAPK7-Thr218/Tyr220 (3371, Cell Signaling), MAPK7 (3372, Cell Signaling), SLC2A1 (sc-7903, Santa Cruz Biotechnology), SLC2A4 (MA183191, Thermo Fisher Scientific), INSR (3025, Cell Signaling), pIRS1-Tyr608 (09–432, Millipore), IRS1 (2382, Cell Signaling),

pCREB1-Ser133 (9198, Cell Signaling), CREB1 (9197, Cell Signaling), CEBPβ (3087, Cell Signaling), pAKT1-Ser473 (9271, Cell Signaling), AKT (9272, Cell Signaling), pMEF2A-S488 (9737, Cell Signaling), MEF2A (ab32866, Abcam), pMEF2C (ab64644, Abcam), MEF2C (SAB4504712, Sigma), cleaved caspase 3 (9661, Cell Signaling), caspase 3 (9661, Cell Signaling) and GB (sc-166123, Santa Cruz Biotechnology). Secondary anti-mouse (7076, Cell Signaling) and anti-rabbit (7074, Cell Signaling) HRP conjugates were used along with the Amersham ECL Prime and Select detection reagents (RPN2232 and RPN2235, Amersham) to detect the bands of interest. Images were obtained with a ChemiDoc MP System (BioRad).

## RNA-Seq and bioinformatics analysis

Total RNA was obtained from whole tissue samples from the non-infarct area (1–3 mm adjacent to infarct) of the hearts from C57BL/6N mice subject to 4 weeks Sham or MI, with two individuals per group. The quality and integrity of total RNA samples were checked using a 2100 Bioanalyzer (Agilent Technologies). RNA-seq libraries were then generated using the TruSeq Stranded mRNA assay (Illumina) according to the manufacturer's instructions. Unmapped paired-reads of 76 bp were interrogated using a quality control pipeline comprising of FastQC v0.11.3 (Babraham Institute 2010) and FastQ Screen v0.9.2 (Babraham Institute 2011). The reads were trimmed to remove any adapter or poor quality sequence using Trimmomatic v0.36 (*Bolger et al., 2014*). Furthermore, the reads were truncated at a sliding 4 bp window with a mean quality <Q20. Afterward, filtered reads were mapped to the mouse reference sequence analysis set (mm10/Dec. 2011/GRCm38) from the UCSC browser using STAR v2.5.3a (*Dobin et al., 2013*). The genome index was created using the Mouse Gencode M16 gene annotation. Properly mapped reads were counted using htseq-count v0.6.1p1. Normalization and differential expression analysis was performed using DESeq2v1.10.1 on Rv3.2.3 (http://www.R-project.org/) (*Love et al., 2014*). Glucose metabolic factors were selected for comparison between Sham and MI groups. The genes with significantly different expression were determined by false discovery rate errors of less than 0.05 for the absolute value of a $\log_2$ fold change.

## Quantitative real-time polymerase reaction (qPCR)

Total RNA from cells and non infarcted ventricular tissue (1–3 mm adjacent to the infarct) was obtained by Trizol extraction. Samples were treated with DNAse (DNA-free Removal Kit, Invitrogen) to eliminate genomic DNA contamination. RNA was converted to cDNA using Oligo(dT) primers (C110A, Promega) and Superscript II Reverse Transcriptase (18064, Invitrogen). Specific primers for quantitative real-time polymerase reaction (qPCR) were purchased from Qiagen and reactions were conducted using SYBR Select PCR Master Mix (4472908, Applied Biosystems) following the manufacturer's instructions. *Pri-mir128-1* and *Pri-mir128-2* primers were purchased from Applied Biosystems and reactions were carried out using TaqMan Gene Expression Master Mix (4369016, Applied Biosystems). qPCR reactions were monitored in the Step One Plus PCR System (Applied Biosystems), and the fold change was calculated by the $2^{-\Delta\Delta Ct}$ method (*Livak and Schmittgen, 2001*). The level of mRNA expression was normalized to *18S* expression.

## Mature miRNA measurement

Mature *mir128-3p* level was gained using TaqMan Advanced miRNA assay (477892_mir, Applied Biosystems) following the manufacturer's instructions. Total RNA was extracted using Trizol. Next, miRNA was converted to cDNA using TaqMan Advanced miRNA cDNA Synthesis Kit (A28007, Applied Biosystems). qPCR reactions were performed using TaqMan Fast Advanced Master Mix (4444557, Applied Biosystems) and monitored in the Step One Plus PCR System (Applied Biosystems) selecting the fast cycling mode. The fold change was calculated by the $2^{-\Delta\Delta Ct}$ method normalizing against *mir191-5p* expression (477952_mir).

## Chromatin Immuno-precipitation

Chromatin immunoprecipitation (ChIP) was performed using the SimpleChip Plus Enzymatic ChIP Kit (9004, Cell Signaling) following the manufacturer's instructions. In brief, approximately $2 \times 10^6$ ARCM cells were cross-linked with 1% formaldehyde and harvested for the preparation of the nuclear suspension. Nuclear membrane was lysed by sonication. Fragmented chromatin was

immunoprecipitated by anti-CEBPβ antibody (sc-7962, Santa Cruz Biotechnology). qPCR was subsequently performed using the primer sets detailed below. Data was normalized to input chromatin.

| Base pairs from TSS | | 5′ to 3′ |
|---|---|---|
| −429 to −259 | F | AGACCCTGTCTTGAAAAACCAAAA |
| | R | TGAGCATGTCAGAATTTGGGAGG |
| −1,361 to −1,214 | F | TCTTTCTTGAGTTAGTGCAGGTAGT |
| | R | CACTGAGCAACTGAACAAGTCT |
| −2,104 to −1,913 | F | GCAGATGGACATGCTGACATACA |
| | R | AGCCTTGACAATAATGAAGCTGTCT |
| −2,550 to −2,383 | F | AGCTACCTTTCCAACCTCTCAA |
| | R | TCAGACAAGATACTCTGTGAGTTGT |
| −3,528 to −3,383 | F | AAAGAAAGGAAAGGAGAAGAAAAGA |
| | R | GATGAAGAAGGCATTTAAACCTTGA |
| −4,047 to −3,873 | F | GTGGACTGCTTCACATGGTT |
| | R | CCGCAAGCCTGCTAATTTAC |
| −4,176 to −4,026 | F | TGAGTAAGTGGATGAATGTTGCT |
| | R | GGAACCATGTGAAGCAGTCC |
| −4,524 to −4,357 | F | GGGAGGTTATTTATCAAAAGTACCA |
| | R | CCCTGCAAAATATACAATTAGCCT |

## Immunoprecipitation

To investigate protein association, immunoprecipitations were performed with Protein G sepharose (Sigma) following the manufacturer's instruction. Briefly, ARCMs were lysed with immunoprecipitation buffer (50 mM Tris, 250 mM NaCl, 0.25% v/v TritonX- 100, and 10% Glycerol, pH7.4). After, quantification, 2 mg of the protein extract was immunoprecipitated with antibodies against CEBPβ (sc-7962, Santa Cruz Biotechnology) or IgG (Cell Signaling, 2729). Immune complexes were eluted in Laemmli sample buffer (65.8 mM Tris-HCl, pH6.8, 2.1% SDS, 26.3% glycerol, 0.01% bromophenol blue). Precipitated and input proteins were subjected to SDS-PAGE and immunoblotted using the respective antibodies.

## Immunofluorescence

To observe nuclear localization, transcriptional factors were detected using immunofluorescence. ARCMs and iPSC-CMs cultured on glass coverslips were fixed with 4% PFA for 15 min at room temperature, followed by permeabilization with 0.1% Triton X + 0.1% sodium citrate for 8 min. Coverslips were later incubated for 1 hr in 3% normal donkey serum (NDS). Primary pCREB1-Ser133 (9198, Cell Signaling) and CEBPβ (3087, Cell Signaling) antibodies were diluted 1:100 in 1% NDS and applied to coverslips for incubation overnight at 4°C. A secondary donkey anti-rabbit-Alexa-488 antibody (711-545-152, Jackson ImmunoResearch) was used after washing with PBS. Finally, the coverslips were washed and mounted with DAPI-Vectashield (H-1200, Vector Laboratories). pCREB1 and CEBPβ nuclear intensity images were collected on a Leica TCS SP8 AOBS upright confocal microscope. For quantification, ten fields (40–60 cells in total) were obtained for each group and image were analyzed with ImageJ.

## SLC2A4 translocation

For examination of SLC2A4 translocation, H9C2 cells were co-transfected with pLenti-myc-*Slc2a4*-mCherry (ID #64049, Addgene), and scramble siRNA or rat *Mapk7* siRNA using Lipofectamine 2000. The resulting overexpressed SLC2A4 protein is fused to an mCherry reporter on its C-terminal, allowing for the detection of total expression, and a c-MYC tag sequence in its N-terminal to detect the transporters translocated to the membrane. After transfection, cells were incubated for 48 hr and serum starved overnight, followed by hypoxic treatment. During the last 30 mins of hypoxia, the

cells were stimulated with 0.1 µmol/L insulin. Afterward, they were fixed with 4% PFA for 15 mins at room temperature. In non-permeabilized condition, cells were labeled with primary anti-MYC antibody (2272, Cell Signaling) overnight at 4° C, followed by anti-rabbit-488 conjugate (711-545-152, Jackson ImmunoResearch) for 2 hr at room temperature. Samples were mounted with DAPI-containing Vectashield (H-1200, Vector Laboratories) before imaging. A Leica TCS SP8 AOBS upright confocal was used to obtain fluorescent images through specific detectors (Texas Red 602–665 nm, FITC 494–530 nm, and DAPI 417–477 nm). When it was not possible to eliminate cross-talk between channels, the images were collected sequentially.

## Glucose uptake

Glucose uptake in cultured cardiomyocytes was measured by the accumulation of 2-deoxy-D-glucose (2DG) using the Glucose Uptake-Glo Assay (J1342, Promega). Following hypoxia, cells were stimulated with 0.1 µmol/L of insulin for 30 mins in no glucose-DMEM (A14430, Gibco). 2DG (1 mM) was added to the medium and further incubated for 20 mins. The insulin and 2DG incubations were maintained in hypoxic conditions. Glucose uptake was stopped and cells were collected to conduct the assay. Luminescence was measured with a micro-plate reader (Gen5, Biotek) with 0.3–1 s integration.

## ATP assay

For ATP measurement, ARCMs were starved during hypoxia with a modified Tyrode solution (120 mmol/L NaCl, 3.3 mmol/L KCl, 24 mmol/L NaHCO$_3$, 1.2 mmol/L KH$_2$PO$_4$, 1 mmol/L CaCl$_2$, 1 mmol/L MgSO$_4$, 0.2% BSA, 1.4 mmol/L glucose, pH7.4). To extract ATP, cells were homogenized with 0.6 mol/L perchloric acid (PCA) and centrifuged for 5 mins at 1000xg. The samples were neutralized with 2 mol/L KOH. PCA was removed by centrifuging for 5 mins at 8,000 g. Samples were diluted and ATP quantified with ATP Bioluminiscent Assay Kit (FLAA, Sigma-Aldrich), following the manufacturer's instructions. Luminescence was measured with a micro-plate reader (Gen5, Biotek).

## Luciferase reporter assay

To determine the effects of hypoxia on *Pri-mir128-1* initiation, the luciferase reporter plasmids (*R3hdm1*-luc containing −1.5 kbp of transcription start site downstream of human *R3hdm1* exon1 or *Pri-mir128-1*-luc containing −1.5 kbp of intronic TSS localized between exon1 and exon2 of *R3hdm1* gene) were constructed by sub-cloning into the pGL3-Basic luciferase plasmid using KpnI and XhoI restriction sites. The luciferase reporters and renilla plasmids were co-transfected into H9C2 and incubated for 48 hr before starting the hypoxic treatment. Cells were collected and luciferase activity measured using Dual-Luciferase Reporter Assay System (E1980, Promega). The pGL3-Basic plasmid without a promoter region was used as a control.

Secondly, to assess the effects of *mir128-3p* on *IRS1* 3'UTR region, a luciferase reporter assay was constructed within pmiRGlo vector (a gift from Delvac Oceandy's lab, University of Manchester). A 500 bp region of human Irs1 3′UTR was cloned into 3′ multiple cloning site (MCS) of the luciferase vector employing NheI and SalI restriction sites. To confirm binding of *mir128-3p* on *IRS1* 3'UTR, four base pairs of the seed sequence were mutated with QuickChange Site-Directed Mutagenesis Kit (200519–5, Agilent Technologies). For the assays, the constructed luciferase vectors were co-transfected with mimic or antagomiR before measuring luciferase activity using the Dual-Luciferase Reporter Assay System. The pmiRGlo vector without any added 3'UTR region was used as a control.

Finally, to evaluate the effects of *mir128-3p* TuD, a reporter plasmid was constructed within pmiRGlo vector. Four consecutive sequences complementary to *mir128-3p* sequence were cloned into 3' MCS of the vector employing SacI and XhoI restriction sites (see below). The forward and reverse strands were synthesized by Sigma. *mir128-3p* activity reporter sequence: GGGAGCTCAAAAGAGACCGGTTCACTGTGAGCAAAGAGACCGGTTCACTGTGAGCAAAGAGACCGGTTCACTGTGAGCAAAGAGACCGGTTCACTGTGACTCGAGGG.

## Data analysis

Data are presented as mean ± SEM and was analyzed using one-way or two-way ANOVA followed by Bonferroni post hoc tests where appropriate. Comparisons between two groups were performed

using Student's t-test. Statistical analysis was performed using the GraphPad Prism eight software. p-values<0.05 were considered statistically significant.

## Acknowledgements

We thank Andy Hayes (Genomic technologies facility, University of Manchester) and Ian Donaldson (Bioinformatics facility, University of Manchester) for technical support on RNA Sequencing performance and analyses. We also thank Roger Meadows and Steven Marsden (Bioimaging facility, University of Manchester) for technical assistance for acquiring confocal images.

This work was supported by the British Heart Foundation (FS/15/16/31477 and FS/18/73/33973 to W Liu and PG/12/76/29852, PG/14/71/31063, PG/14/70/31039 to X Wang) and a professorship from the German Centre for Cardiovascular Research (81Z0700201 to O J Müller).

## Additional information

### Funding

| Funder | Grant reference number | Author |
| --- | --- | --- |
| British Heart Foundation | FS/15/16/31477 | Wei Liu |
| British Heart Foundation | FS/18/73/33973 | Wei Liu |
| British Heart Foundation | PG/12/76/29852 | Xin Wang |
| British Heart Foundation | PG/14/71/31063 | Xin Wang |
| British Heart Foundation | PG/14/70/31039 | Xin Wang |
| German Centre for Cardiovascular Research | 81Z0700201 | Oliver J Müller |

The funders had no role in study design, data collection and interpretation, or the decision to submit the work for publication.

### Author contributions

Andrea Ruiz-Velasco, Conceptualization, Formal analysis, Investigation, Visualization, Methodology, Writing - original draft; Min Zi, Investigation, Methodology; Susanne S Hille, Karolina Sekeres, Resources, Methodology; Tayyiba Azam, Namrita Kaur, Juwei Jiang, Binh Nguyen, Pablo Binder, Lucy Collins, Investigation; Fay Pu, Norbert Frey, Writing - review and editing; Han Xiao, Elizabeth J Cartwright, Methodology, Writing - review and editing; Kaomei Guan, Validation, Writing - review and editing; Oliver J Müller, Conceptualization, Funding acquisition, Methodology, Writing - original draft; Xin Wang, Conceptualization, Supervision, Funding acquisition, Writing - original draft; Wei Liu, Conceptualization, Formal analysis, Supervision, Funding acquisition, Investigation, Writing - original draft, Project administration

### Author ORCIDs

Wei Liu https://orcid.org/0000-0003-1592-6693

### Ethics

Animal experimentation: All animal studies were performed in accordance with the United Kingdom Animals (Scientific Procedures) Act 1986 under the Home Office license P3A97F3D1, and were approved by the University of Manchester Ethics Committee.

### Decision letter and Author response

Decision letter https://doi.org/10.7554/eLife.54298.sa1
Author response https://doi.org/10.7554/eLife.54298.sa2

## Additional files

### Supplementary files
- Supplementary file 1. Key resources table.
- Transparent reporting form

### Data availability
All data generated or analysed during this study are included in the manuscript and supporting files. Source data files have been provided for all figures.

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
