## [Decision Letter]

**Acceptance summary:**

We find that your study highlights the important role that microRNAs perform in the adaptive response of the heart to acute injury. Moreover, the finding that miR128-3 could be targeted to ameliorate post-MI cardiac remodeling may provide a new therapeutic approach to heart failure.

**Decision letter after peer review:**

Thank you for submitting your article "Targeting miR128-3p alleviates myocardial insulin resistance and prevents ischemia-induced heart failure" for consideration by *eLife*. Your article has been reviewed by three peer reviewers, including Milica Radisic as the Reviewing Editor and Reviewer #1, and the evaluation has been overseen by Edward Morrisey as the Senior Editor. The following individuals involved in review of your submission have agreed to reveal their identity: Ruiping Xiao (Reviewer #2); Tingsen Benson Lim (Reviewer #3).

The reviewers have discussed the reviews with one another and the Reviewing Editor has drafted this decision to help you prepare a revised submission.

Summary:

This is a comprehensive and a very thorough manuscript focusing on elucidating reasons that underlie insulin resistance post myocardial ischemia such as MI. The authors used post-MI mouse hearts, neonatal and adult rat cardiomyocytes as well as human iPSC derived cardiomyocytes to demonstrate the central role of Insulin Receptor Substrate 1 downregulation by a novel micro RNA, miR128-3p. They thoroughly investigated both upstream events and downstream consequences that result in lower availability of GLUT4 transporter on the cardiomyocyte surface that overall decreases glucose uptake after ischemia and fails to support mostly glycolytic metabolism in these cardiomyocytes

Essential revisions:

Addressing the below comments will enable the authors to improve the manuscript further:

1) The authors should provide more evidence on the causal relationship between the upregulation of miR128-3p and downregulation of IRS-1 in response to ischemic injury. In Figure 1J, IRS-1 mRNA was decreased 6 h after hypoxia in cardiomyocytes. If the reduced IRS-1 was caused by miR128-3p, miR128-3p should increase before that. However, in Figure 2E, the authors only show miR128-3p was increased 6 h after hypoxia. An earlier time point should be investigated.

2) Although most of the data are very solid quite consistent. Some data in the manuscript showed internal inconsistence. In Figures 5B and 6B, the AAV-GFP group exhibited significant decreased cardiac function 1 wk after myocardial infarction. However, in Figure 4B, there was no change of the Flox group after the same period after infarction. Please explain.

3) The infarcted hearts are of great heterogeneity. Which part of the hearts was analyzed for insulin sensitivity and the related signaling, ischemic, adjacent or remote zone? Why? Please specify.

4) Immunofluorescence staining is not necessarily ideal quantitative measurement of CREB and CEBPβ in nuclei (Figure 3M and Figure 3—figure supplement 7). Fluorescence staining can be subjective and with a measurement of just 5 nuclei is clearly insufficient and inconclusive. Also, background noise can be observed for both CREB and CEBPβ staining. An immunoblot, which the authors have duly performed (Figure 3O), will suffice. However, an appropriate control with siScramble is missing with the experiments performed on Figure 3O.

5) Relative heart weight and CM size were significantly increased 1-w post MI (Figure 4—figure supplement 1B and C) but unchanged at the same condition in Figure 5—figure supplement 1. Can the authors explained this discrepancy?

6) The authors showed when Erk5-cko was rescued with AAV9-ERK5. However, the level of ERK5 was restored to a physiological level close to but not at a higher level to Erk-flox control. This piece of data is important because, physiologically, the Erk5-cko AAV9-ERK5 mice would be akin to a wildtype or Erk-flox control and would respond to MI as a wildtype status. However, the data in Figure 5 showed attenuation of MI in the Erk5-cko AAV9-ERK5 mice but made no comparison to the Erk5-flox control which may be important.

Reviewer #1:

This is a comprehensive and a very thorough manuscript focusing on elucidating reasons that underlie insulin resistance post myocardial ischemia such as MI. The authors used post-MI mouse hearts, neonatal and adult rat cardiomyocytes as well as human iPSC derived cardiomyocytes to demonstrate the central role of Insulin Receptor Substrate 1 downregulation by a novel micro RNA, miR128-3p. They thoroughly investigated both upstream events and downstream consequences that result in lower availability of GLUT4 transporter on the cardiomyocyte surface that overall decreases glucose uptake after ischemia and fails to support mostly glycolytic metabolism in these cardiomyocytes.

Briefly, the authors elucidated the central role of ERK5 that leads to activation of CREB followed by activation of CEBPβ that then block activity of miR128-3p which enables IRS1 expression and sufficient glucose utilization. Under the conditions of prolonged hypoxia, ERK5 is lost leading to miR128-3p activation and ultimately IRS1 loss. This new knowledge could potentially lead to the development of new therapies, when applied in the right window post-MI, through e.g. application of AntimiR128 as the authors have shown here. However, ultimately re-establishment of healthy lipid metabolism in these CMs might be the most desired target.

Overall, in the spirit of *eLife* which is not to request un-necessary revisions, I think this is a pretty complete story which is suitable for publication.

Reviewer #2:

In the current manuscript, Ruiz-Velasco et al. provided compelling evidence that have been generated with state-of-the-art techniques on the role of miR128-3p-mediated cardiac insulin resistance in myocardial ischemic injury. They show that ERK5/CEBPβ/ miR128-3p was enhanced in response to cardiac ischemic insult, which subsequently down-regulated IRS-1 and induced insulin resistance. The findings are novel, important, and of great clinical relevance.

I have the following concerns.

1) The authors should provide more evidence on the causal relationship between the upregulation of miR128-3p and downregulation of IRS-1 in response to ischemic injury. In Figure 1J, IRS-1 mRNA was decreased 6 h after hypoxia in cardiomyocytes. If the reduced IRS-1 was caused by miR128-3p, miR128-3p should increase before that. However, in Figure 2E, the authors only show miR128-3p was increased 6 h after hypoxia. An earlier time point should be investigated.

2) Although most of the data are very solid quite consistent. Some data in the manuscript showed internal inconsistence. In Figures 5B and 6B, the AAV-GFP group exhibited significant decreased cardiac function 1 wk after myocardial infarction. However, in Figure 4B, there was no change of the Flox group after the same period after infarction. Please explain.

3) The infarcted hearts are of great heterogeneity. Which part of the hearts was analyzed for insulin sensitivity and the related signaling, ischemic, adjacent or remote zone? Why? Please specify.

Reviewer #3:

Ruiz-Velasco et al. conducted a study demonstrating IRS1 was reduced in mice suffering MI and that miR-128-3p is elevated in prolong MI. Mechanistically, they demonstrated miR-128-3p targets and degrade Irs1 and that ERK5 mediated CEBPβ transcriptionally represses miR-128-3p under hypoxia. More importantly, they showed AAV9 delivery of ERK5 or CEBPβ or anti-miR-128-3p impeded cardiac injury post-MI in vivo, thereby, concluding that targeting miR-128-3p alleviates cardiac insulin resistance and may impedes the progression of heart failure post-ischemia. They also looked into the human-relevant for the ERK5-CEBPβ signaling pathway negative regulation of miR128-3p in response to hypoxia through iPSC-CMs and showed that the pathway preserves IRS1 expression through repression of miR128-3p suggesting beneficial effects on the insulin dependent pathway in human iPSC-CMs under hypoxic condition. Overall, the authors have done an impressive work on elucidating the ERK5-CEBPβ signaling cascade negative regulation of miR128-3p under ischemia/hypoxic condition. The manuscript is well designed and written with comprehensive data. The results are interesting and potentially important to our understanding on myocardial insulin resistance and ischemia-induced heart failure. However, I do have concerns listed below which I hope the authors are able to address.

1) I do not necessarily agree using immunofluorescence staining as a quantitative measurement of CREB and CEBPβ in nuclei (Figure 3M and Figure 3—figure supplement 7). Fluorescence staining can be subjective and with a measurement of just 5 nuclei is clearly insufficient and inconclusive. Also, background noise can be observed for both CREB and CEBPβ staining. I would suggest to deflect Figure 3M to supplementary or be removed. An immunoblot, which the authors have duly performed (Figure 3O), will suffice. However, an appropriate control with siScramble is missing with the experiments performed on Figure 3O.

2) Relative heart weight and CM size were significantly increased 1-w post MI (Figure 4—figure supplement 1B and C) but unchanged at the same condition in Figure 5—figure supplement 1. Can the authors explained this discrepancy?

3) The authors showed when Erk5-cko was rescued with AAV9-ERK5. However, the level of ERK5 was restored to a physiological level close to but not at a higher level to Erk-flox control. This piece of data is important because, physiologically, the Erk5-cko AAV9-ERK5 mice would be akin to a wildtype or Erk-flox control and would respond to MI as a wildtype status. However, the data in Figure 5 showed attenuation of MI in the Erk5-cko AAV9-ERK5 mice but made no comparison to the Erk5-flox control which I think is important.

---

## [Author Response]

Essential revisions:Addressing the below comments will enable the authors to improve the manuscript further:1) The authors should provide more evidence on the causal relationship between the upregulation of miR128-3p and downregulation of IRS-1 in response to ischemic injury. In Figure 1J, IRS-1 mRNA was decreased 6 h after hypoxia in cardiomyocytes. If the reduced IRS-1 was caused by miR128-3p, miR128-3p should increase before that. However, in Figure 2E, the authors only show miR128-3p was increased 6 h after hypoxia. An earlier time point should be investigated.

We do appreciate the critical consideration. As suggested, we performed various durations of hypoxia on rat cardiomyocytes to determine whether miR128-3p augmentation is a causative factor for the decreased IRS1. Our new data demonstrated that both the transcript of miR128-3p (Pri-miRNA128-1) and mature miR128-3p were rising 4 hours post-hypoxia, and constantly increased after 6 hours of hypoxia, which indicates miR128-3p, at least partially, causes IRS1 degradation. The new data (Figure 2—figure supplement 2 and Figure 2—figure supplement 4) and text have been added in the revised manuscript.

2) Although most of the data are very solid quite consistent. Some data in the manuscript showed internal inconsistence. In Figures 5B and 6B, the AAV-GFP group exhibited significant decreased cardiac function 1 wk after myocardial infarction. However, in Figure 4B, there was no change of the Flox group after the same period after infarction. Please explain.

Sorry about the confusion, and we would like to clarify this misunderstanding. Figure 4B demonstrated that, compared to Sham group, Flox group did not display significant impaired cardiac function yet after MI surgery for 1 week; whereas Erk5-CKO showed decreased function. We believe that Erk5 deficiency is likely one of the causes leading to heart failure after ischemia injury. In both Figure 5 and Figure 6, *all* experiments were conducted on Erk5-CKO mice: after 1 week, decreased function was observed in Erk5-CKO mice with AAV-GFP injection (as injection control), however, either restoration of Erk5 (AAV-Erk5, in Figure 5) or CEBPβ (AAV-CEBPβ, in Figure 6) rescued the heart function in Erk5-CKO.

3) The infarcted hearts are of great heterogeneity. Which part of the hearts was analyzed for insulin sensitivity and the related signaling, ischemic, adjacent or remote zone? Why? Please specify.

We would like to specify that all molecular mechanisms analyses were conducted in the non-infarcted ventricular tissue (1-3 mm adjacent to infarct). The ischemic area contains a majority of dead cells; however, the non-infarcted area is in hypoxia condition due to limited angiogenesis. Our study provides new insights into the therapeutic potential of post-ischemia heart failure by maintaining insulin sensitivity in the myocardium in the hypoxic area. In other words, preserving cardiac insulin sensitivity in hypoxic myocardium is able to reserve the whole heart performance and slow down the progression of heart failure after ischemia injury. The detailed information has been added in the Materials and methods.

4) Immunofluorescence staining is not necessarily ideal quantitative measurement of CREB and CEBPβ in nuclei (Figure 3M and Figure 3—figure supplement 7). Fluorescence staining can be subjective and with a measurement of just 5 nuclei is clearly insufficient and inconclusive. Also, background noise can be observed for both CREB and CEBPβ staining. An immunoblot, which the authors have duly performed (Figure 3O), will suffice. However, an appropriate control with siScramble is missing with the experiments performed on Figure 3O.

We apologize for the mislabelling. For quantification of Figure 3M and Figure 3—figure supplement 7, we randomly selected and analyzed ten fields (40-60 cells in total) for each group per individual experiment. 'N=5' meant 5 individual experiments, not 5 nuclei. We have amended the description in a correct way in the legend. Although there is fluorescent background due to the second antibody, we focused on nuclear staining, because CREB and CEBPβ function as transcription factors when localized in nuclei. We have amended the Y axis in figures to make it clear. In addition, we have provided the control with siScramble only in Figure 3O.

5) Relative heart weight and CM size were significantly increased 1-w post MI (Figure 4—figure supplement 1B and C) but unchanged at the same condition in Figure 5—figure supplement 1. Can the authors explained this discrepancy?

Sorry about the confusion, and we would like to clarify this misunderstanding. Figure 4 demonstrated that, compared to Flox Sham group, Flox hearts displayed increased heart weight and CM size after MI surgery for 1 week; whereas Erk5-CKO did not. However, in Figure 5, *all* experiments were conducted on Erk5-CKO mice: as expected, AAV-GFP injection (as injection control) in Erk5-CKO mice did not show increased heart weight and CM after 1 week MI, however, restoration of Erk5 (AAV-Erk5) resulted in the hypertrophic growth, akin to the observation in Flox-MI hearts.

6) The authors showed when Erk5-cko was rescued with AAV9-ERK5. However, the level of ERK5 was restored to a physiological level close to but not at a higher level to Erk-flox control. This piece of data is important because, physiologically, the Erk5-cko AAV9-ERK5 mice would be akin to a wildtype or Erk-flox control and would respond to MI as a wildtype status. However, the data in Figure 5 showed attenuation of MI in the Erk5-cko AAV9-ERK5 mice but made no comparison to the Erk5-flox control which may be important.

Thanks for the scientific comments. We compared the Flox-MI group with AAV-Erk5 injected Erk5-CKO, and found that restoring ERK5 at a physiological level reduced the infarction to a similar level observed in Erk5-flox hearts. We have added the result in Figure 5—figure supplement 2 and the text accordingly.

Reviewer #1:This is a comprehensive and a very thorough manuscript focusing on elucidating reasons that underlie insulin resistance post myocardial ischemia such as MI. The authors used post-MI mouse hearts, neonatal and adult rat cardiomyocytes as well as human iPSC derived cardiomyocytes to demonstrate the central role of Insulin Receptor Substrate 1 downregulation by a novel micro RNA, miR128-3p. They thoroughly investigated both upstream events and downstream consequences that result in lower availability of GLUT4 transporter on the cardiomyocyte surface that overall decreases glucose uptake after ischemia and fails to support mostly glycolytic metabolism in these cardiomyocytes.Briefly, the authors elucidated the central role of ERK5 that leads to activation of CREB followed by activation of CEBPβ that then block activity of miR128-3p which enables IRS1 expression and sufficient glucose utilization. Under the conditions of prolonged hypoxia, ERK5 is lost leading to miR128-3p activation and ultimately IRS1 loss. This new knowledge could potentially lead to the development of new therapies, when applied in the right window post-MI, through e.g. application of AntimiR128 as the authors have shown here. However, ultimately re-establishment of healthy lipid metabolism in these CMs might be the most desired target.

Thanks for the scientific comments. As we know, fatty acid (70%) and glucose (30%) are the two main source of energy in the cardiomyocytes, both of which are essential for maintaining heart function in response to various physiological and pathological stresses. In the situation of hypoxia, glucose provides a small amount of ATP by glycolysis in the myocardium, which becomes a paramount compensatory energy source during short-term oxygen deprivation. We completely agree with the reviewer's comments that modulating fatty acid utilization will also be a pivotal way to preserve cardiac function. Actually, our RNA-Seq data also detected the abnormalities of genes participating in fatty acid metabolism. However, the current study is focusing on the investigation of importance of cardiac insulin sensitivity during short-term ischemic stress. We do not exclude the significance of fatty acid metabolism in such a condition, instead, we believe that keeping the balance of fatty acid and glucose metabolism in the fact of disease for various durations is vital for heart performance. The more comprehensive understanding of either fatty acid or glucose utilization will be explored in future studies.

Overall, in the spirit of eLife which is not to request un-necessary revisions, I think this is a pretty complete story which is suitable for publication.Reviewer #2:[…] I have the following concerns.1) The authors should provide more evidence on the causal relationship between the upregulation of miR128-3p and downregulation of IRS-1 in response to ischemic injury. In Figure 1J, IRS-1 mRNA was decreased 6 h after hypoxia in cardiomyocytes. If the reduced IRS-1 was caused by miR128-3p, miR128-3p should increase before that. However, in Figure 2E, the authors only show miR128-3p was increased 6 h after hypoxia. An earlier time point should be investigated.

We do appreciate the critical consideration. As suggested, we performed various durations of hypoxia on rat cardiomyocytes to determine whether miR128-3p augmentation is a causative factor for the decreased IRS1. Our new data demonstrated that both the transcript of miR128-3p (Pri-miRNA128-1) and mature miR128-3p were rising 4 hours post-hypoxia, and constantly increased after 6 hours of hypoxia, which indicates miR128-3p, at least partially, causes IRS1 degradation. The new data (Figure 2—figure supplement 2 and Figure 2—figure supplement 4) and text have been added in the revised manuscript.

2) Although most of the data are very solid quite consistent. Some data in the manuscript showed internal inconsistence. In Figures 5B and 6B, the AAV-GFP group exhibited significant decreased cardiac function 1 wk after myocardial infarction. However, in Figure 4B, there was no change of the Flox group after the same period after infarction. Please explain.

Sorry about the confusion, and we would like to clarify this misunderstanding. Figure 4B demonstrated that, compared to Sham group, Flox group did not display significant impaired cardiac function yet after MI surgery for 1 week; whereas Erk5-CKO showed decreased function. We believe that Erk5 deficiency is likely one of the causes leading to heart failure after ischemia injury. In both Figure 5 and Figure 6, *all* experiments were conducted on Erk5-CKO mice: after 1 week, decreased function was observed in Erk5-CKO mice with AAV-GFP injection (as injection control), however, either restoration of Erk5 (AAV-Erk5, in Figure 5) or CEBPβ (AAV-CEBPβ, in Figure 6) rescued the heart function in Erk5-CKO.

3) The infarcted hearts are of great heterogeneity. Which part of the hearts was analyzed for insulin sensitivity and the related signaling, ischemic, adjacent or remote zone? Why? Please specify.

We would like to specify that all molecular mechanisms analyses were conducted in the non-infarcted ventricular tissue (1-3 mm adjacent to infarct). The ischemic area contains a majority of dead cells; however, the non-infarcted area is in hypoxia condition due to limited angiogenesis. Our study provides new insights into the therapeutic potential of post-ischemia heart failure by maintaining insulin sensitivity in the myocardium in the hypoxic area. In other words, preserving cardiac insulin sensitivity in hypoxic myocardium is able to reserve the whole heart performance and slow down the progression of heart failure after ischemia injury. The detailed information has been added in the Materials and methods.

Reviewer #3:[…] I do have concerns listed below which I hope the authors are able to address.1) I do not necessarily agree using immunofluorescence staining as a quantitative measurement of CREB and CEBPβ in nuclei (Figure 3M and Figure 3—figure supplement 7). Fluorescence staining can be subjective and with a measurement of just 5 nuclei is clearly insufficient and inconclusive. Also, background noise can be observed for both CREB and CEBPβ staining. I would suggest to deflect Figure 3M to supplementary or be removed. An immunoblot, which the authors have duly performed (Figure 3O), will suffice. However, an appropriate control with siScramble is missing with the experiments performed on Figure 3O.

We apologize for the mislabelling. For quantification of Figure 3M and Figure 3—figure supplement 7, we randomly selected and analyzed ten fields (40-60 cells in total) for each group per individual experiment. 'N=5' meant 5 individual experiments, not 5 nuclei. We have amended the description in a correct way in the legend. Although there is fluorescent background due to the second antibody, we focused on nuclear staining, because CREB and CEBPβ function as transcription factors when localized in nuclei. We have amended the Y axis in figures to make it clear. In addition, we have provided the control with siScramble only in Figure 3O.

2) Relative heart weight and CM size were significantly increased 1-w post MI (Figure 4—figure supplement 1B and C) but unchanged at the same condition in Figure 5—figure supplement 1. Can the authors explained this discrepancy?

Sorry about the confusion, and we would like to clarify this misunderstanding. Figure 4 demonstrated that, compared to Flox Sham group, Flox hearts displayed increased heart weight and CM size after MI surgery for 1 week; whereas Erk5-CKO did not. However, in Figure 5, *all* experiments were conducted on Erk5-CKO mice: as expected, AAV-GFP injection (as injection control) in Erk5-CKO mice did not show increased heart weight and CM after 1 week MI, however, restoration of Erk5 (AAV-Erk5) resulted in the hypertrophic growth, akin to the observation in Flox-MI hearts.

3) The authors showed when Erk5-cko was rescued with AAV9-ERK5. However, the level of ERK5 was restored to a physiological level close to but not at a higher level to Erk-flox control. This piece of data is important because, physiologically, the Erk5-cko AAV9-ERK5 mice would be akin to a wildtype or Erk-flox control and would respond to MI as a wildtype status. However, the data in Figure 5 showed attenuation of MI in the Erk5-cko AAV9-ERK5 mice but made no comparison to the Erk5-flox control which I think is important.

Thanks for the scientific comments. We compared the Flox-MI group with AAV-Erk5 injected Erk5-CKO, and found that the retrieved ERK5 at a physiological level reduced the infarction to a similar level observed in Erk5-flox hearts. We have added the result in Figure 5—figure supplement 2 and the text accordingly.